# Functional and dynamic polymerization of the ALS-linked protein TDP-43 antagonizes its pathologic aggregation

Tariq Afroz[1], Eva-Maria Hock[1], Patrick Ernst[2], Chiara Foglieni[3], Melanie Jambeau[1], Larissa A.B. Gilhespy[1], Florent Laferriere[1], Zuzanna Maniecka[1], Andreas Plückthun[2], Peer Mittl[2], Paolo Paganetti[3], Frédéric H.T. Allain[4] & Magdalini Polymenidou[1]

TDP-43 is a primarily nuclear RNA-binding protein, whose abnormal phosphorylation and cytoplasmic aggregation characterizes affected neurons in patients with amyotrophic lateral sclerosis and frontotemporal dementia. Here, we report that physiological nuclear TDP-43 in mouse and human brain forms homo-oligomers that are resistant to cellular stress. Physiological TDP-43 oligomerization is mediated by its N-terminal domain, which can adopt dynamic, solenoid-like structures, as revealed by a 2.1 Å crystal structure in combination with nuclear magnetic resonance spectroscopy and electron microscopy. These head-to-tail TDP-43 oligomers are unique among known RNA-binding proteins and represent the functional form of the protein in vivo, since their destabilization results in loss of alternative splicing regulation of known neuronal RNA targets. Our findings indicate that N-terminal domain-driven oligomerization spatially separates the adjoining highly aggregation-prone, C-terminal low-complexity domains of consecutive TDP-43 monomers, thereby preventing low-complexity domain inter-molecular interactions and antagonizing the formation of pathologic aggregates.

[1] Institute of Molecular Life Sciences, University of Zurich, Winterthurerstrasse 190, CH-8057 Zurich, Switzerland. [2] Department of Biochemistry, University of Zurich, Winterthurerstrasse 190, CH-8057 Zurich, Switzerland. [3] Laboratory for Biomedical Neurosciences, Neurocenter of Southern Switzerland, Via Tesserete 46, CH-6900 Lugano, Switzerland. [4] Institute of Molecular Biology and Biophysics, ETH Zurich, CH-8093 Zurich, Switzerland. Eva-Maria Hock and Patrick Ernst contributed equally to this work. Correspondence and requests for materials should be addressed to M.P. (email: magdalini.polymenidou@imls.uzh.ch)

Amyotrophic lateral sclerosis (ALS) and frontotemporal dementia (FTD) are two intimately related adult-onset neurodegenerative disorders[1]. In ALS the loss of motor neurons leads to fatal paralysis typically within 1–5 years of onset, whereas FTD causes severe atrophy of the frontal and temporal lobes leading to behavioral and language dysfunction[2–5]. The RNA-binding protein (RBP) TDP-43 (Trans-activation response element (TAR) DNA-binding protein 43) is mislocalized and aggregated into neuronal cytoplasmic ubiquitinated inclusions in the vast majority of ALS, as well as approximately half of FTD cases[2]. Even though the loss of normal nuclear localization and cytoplasmic TDP-43 aggregation correlates with neurodegeneration, the exact mechanisms of neurotoxicity remain elusive. Moreover, the molecular mechanisms triggering TDP-43 pathology in ALS and FTD remain poorly understood, in part due to lack of high-resolution structural information of TDP-43 in the physiological and pathological state.

TDP-43 is a nucleo-cytoplasmic shuttling protein comprised of two RNA recognition motifs (RRMs) that bind TG-/UG-repeat nucleic acids in a sequence-specific fashion and are indispensable for its roles in RNA metabolism[3, 6–8]. TDP-43 also contains a C-terminal prion-like or low-complexity domain (LCD)[9], mediating protein–protein interactions[10] and also its incorporation into stress granules[11], potentially via its property to phase separate[12, 13]. Furthermore, the TDP-43 LCD is crucially involved in disease, since it is proteolytically cleaved and abnormally phosphorylated, leading to its cytoplasmic accumulation in complex with the full-length protein[2, 14]. Finally, TDP-43 contains an N-terminal region spanning its first 80 amino acids, whose role in the function and/or malfunction of the protein remains unclear. In fact, due to the lack of sequence homology of this region with any known structures, the N-terminal domain (NTD) of TDP-43 was initially thought to be unstructured. However, it was recently reported that monomeric TDP-43 NTD can adopt a Ubiquitin-like[15] or DIshevelled and aXin (DIX)-domain-like[16] fold in solution. Intriguingly, while several recent studies highlighted the importance of NTD for functional TDP-43 dimerization[17] and nucleic acid interaction[15, 18–21], others argued that the same domain promoted pathologic cytoplasmic aggregation and neurotoxicity[20, 22, 23]. Moreover, a small fraction of TDP-43 was reported to exist as dimers in cells, which led to speculation that TDP-43 dimers may initiate or "seed" the formation of high molecular weight pathologic TDP-43 aggregates[17].

Here, we show that physiological TDP-43 exists as nuclear oligomers that are distinct from cytoplasmic complexes formed upon cellular stress or pathologic aggregates. To elucidate the molecular basis of physiological TDP-43 oligomerization, we determined the crystal structure of TDP-43 NTD at 2.1 Å resolution, which revealed an unprecedented mode of head-to-tail interactions between monomers generating solenoid-like polymers. Consistent with the crystal structure, solution NMR spectroscopy confirmed the dynamic nature of inter-molecular and low micromolar affinity electrostatic interactions that stabilize these polymers. Destabilizing oligomerization by point mutations resulted in loss of TDP-43 regulation of alternative splicing of known neuronal RNA targets, indicating that these dynamic TDP-43 oligomers are the functional form of the protein in vivo. Tripartite GFP complementation experiments in cells illustrate that physiological TDP-43 oligomerization prevents LCD intermolecular interactions. Importantly, we show that NTD-driven TDP-43 oligomerization antagonizes pathologic aggregation. This dynamic head-to-tail polymerization of TDP-43, which is reminiscent of DIX domains[24] involved in Wnt signaling, is unique among RBPs and broadens our understanding of TDP-43 function. Most excitingly, our findings indicate that stabilization of functional TDP-43 oligomers could have therapeutic potential by counteracting pathologic aggregation and restoring nuclear function.

## Results

**TDP-43 forms physiological oligomers in human tissues.** In order to capture the in vivo protein state, we systematically analyzed physiological TDP-43 in cells and tissues by in situ, chemically induced, cross-linking prior to cell/tissue lysis. For this purpose, we used disuccinimidylglutarate (DSG), a membrane-permeable cross-linker, which allows preservation of native protein–protein interactions[25]. We performed DSG cross-linking in normal human fibroblasts and brain tissue, followed by nucleo-cytoplasmic fractionation and immunoblotting, in order to analyze the subcellular localization of TDP-43 oligomers and monomers. Upon increasing the DSG concentration, a specific ladder of slowly migrating TDP-43 complexes is detected, consistent with increasing multimeric species of TDP-43 in human fibroblasts (Fig. 1a, Supplementary Fig. 1a–e) and human brain (Fig. 1b, Supplementary Fig. 1f–g). The slightly different migration pattern of TDP-43 oligomers in human motor cortex (Supplementary Fig. 1g) is due to the different gel (4–12% polyacrylamide) used to allow better resolution of autopsy human brain material, which after cross-linking migrates poorly in the regular 12% polyacrylamide gel (Fig. 1b).The higher molecular weight TDP-43 species correspond to distinct molecular masses of the dimer, trimer, tetramer and higher, indicating that native cellular TDP-43 exists not only as dimers, but rather in a spectrum of oligomeric species. The fraction of TDP-43 present in high molecular weight oligomers is comparable among various samples analyzed (Supplementary Fig. 1h). To further validate the specificity of the DSG cross-linking protocol, we investigated the cross-linking pattern of other hnRNPs such as hnRNPA1 and FUS that harbor similar RNA binding and LCDs as TDP-43, but lack the NTD. For hnRNPA1, we detect a specific dimer band predominantly in the nuclear fractions (Supplementary Fig. 2a). However, no intermediate oligomeric bands are detected as for TDP-43 (Fig. 1a). Interestingly, for both hnRNPA1 and FUS, very high molecular weight complexes that are unable to enter the 4–12% denaturing polyacrylamide gels are detected (Supplementary Fig. 2a). This is consistent to previous reports suggesting oligomerization of hnRNPA1[26] and FUS[27] upon RNA binding. In contrast, wild-type superoxide dismutase 1 (SOD1), which is known to exist exclusively as cytoplasmic homodimers[28], showed only the presence of dimeric protein without any non-specific cross-linking, even at the highest DSG concentration used both in human fibroblasts and mouse brain slices (Supplementary Fig. 2b–c). Multimeric TDP-43 species are detected predominantly in the nucleus, correlating with the normal nuclear localization of TDP-43 in healthy cells (Fig. 1a).

**Nuclear TDP-43 oligomers are resistant to oxidative stress.** TDP-43 is recruited to stress granules upon oxidative stress and forms high molecular weight complexes in non-reducing conditions[29–31]. In order to determine if normal nuclear TDP-43 oligomerization is changed upon cellular stress, we treated mouse cortico-hippocampal organotypic slice cultures with sodium arsenite and then performed in situ cross-linking by DSG followed by nucleo-cytoplasmic fractionation (Fig. 1c). Arsenite induces oxidative stress by altering the redox potential in the cell. Consistent with previously published work on cell lines[29], 2 h of arsenite treatment results in a small fraction of TDP-43 to localize into cytoplasmic TIA-1 positive granules (white arrows in Fig. 1d). In both control and arsenite-treated slices, the majority of oligomeric TDP-43 resides in the nuclear fraction, suggesting that this oligomeric state is distinct from the cytoplasmic TDP-43 pool residing in stress granule (Fig. 1e, Supplementary Fig. 2d–e). Furthermore, nuclear TDP-43 oligomerization is independent of

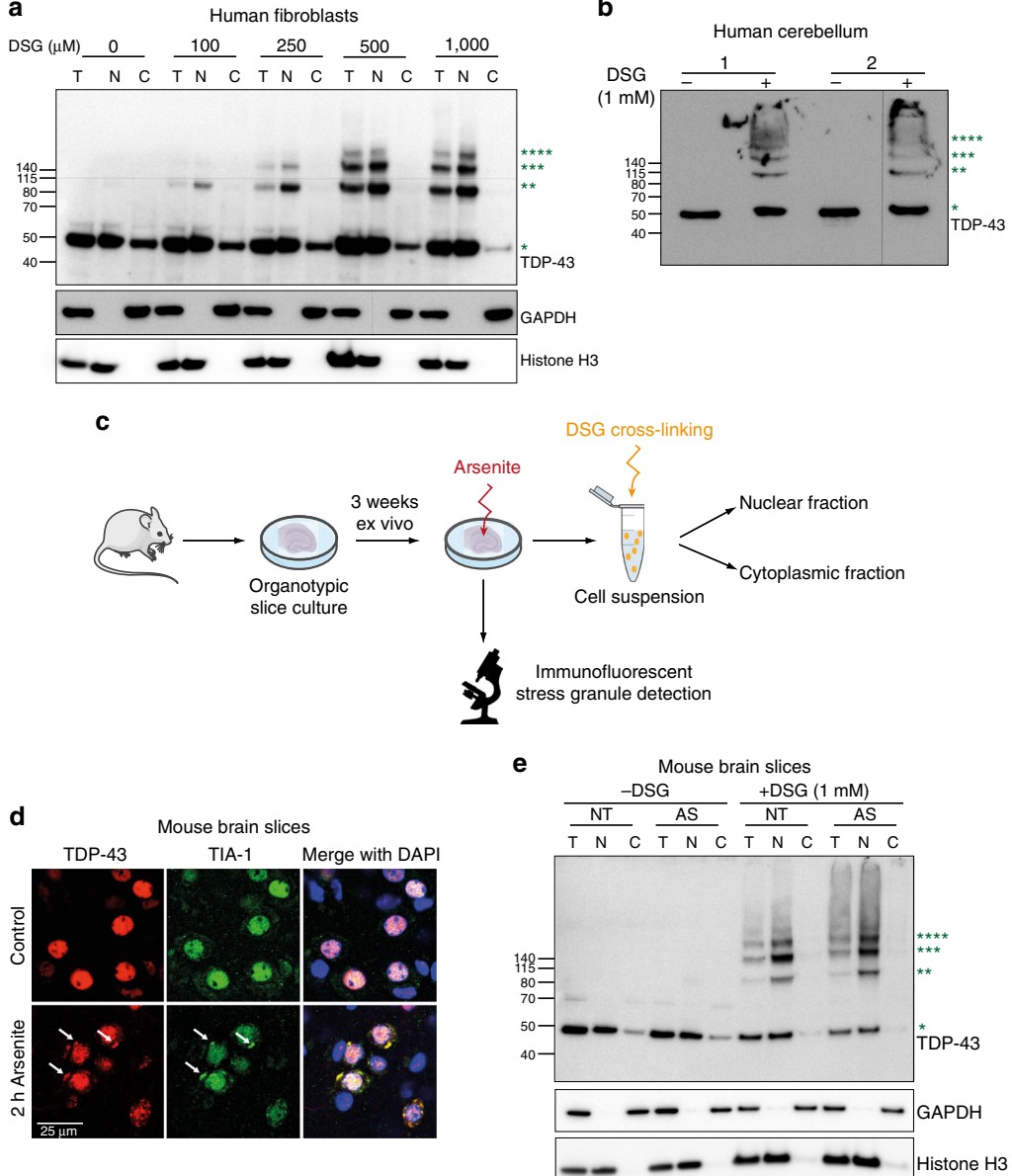

**Fig. 1** TDP-43 oligomers are expressed in human cells and brain and resist oxidative stress. **a** Immunoblots (from 12% denaturing polyacrylamide gels) of human fibroblast fractions obtained upon incubation with increasing concentration of DSG cross-linker followed by nucleo-cytoplasmic fractionation. In the *upper panel*, high molecular weight bands (marked with *green asterisks*) are detected with anti-human TDP-43 antibody at increasing concentrations of DSG in the nuclear fraction (N) and in total cell lysates (T), but not in the cytoplasmic fraction (C). *Lower panels* show immunoblots for cytoplasmic marker GAPDH, or nuclear marker histone H3. Immunoblots are representative of three independent experiments. **b** Immunoblots (from 12% denaturing polyacrylamide gels) of human cerebellum samples. High molecular weight TDP-43 bands are detected in 1 mM DSG cross-linked samples (*green asterisks*) compared to samples obtained in the absence of cross-linker. Immunoblots are representative of three independent experiments. Full immunoblots with molecular weight marker are shown in Supplementary Fig. 1. **c** Schematic diagram depicting the experimental set-up for oxidative stress treatment in mouse organotypic slice cultures followed by DSG mediated cross-linking. Subset of images (such as eppendorf tubes and microscope) in the schematic figure were adapted from the Servier medical art (http://www.servier.com/slidekit). **d** Confocal microscopy images of non-treated (control) and arsenite-treated (2 h Arsenite) mouse brain slices, immunostained with antibodies against TDP-43 and TIA-1. Cytoplasmic TDP-43 granules (*red*) co-localize with the stress-granule marker TIA-1 (*green*). Merged images including DAPI-staining of nuclei (*blue*) are shown in the *right panels*. **e** Immunoblots (from 12% denaturing polyacrylamide gels) of control (NT) and arsenite-treated (As) mouse brain slices after cross-linking by DSG and nucleo-cytoplasmic fractionation. DSG-cross-linked oligomeric TDP-43 (marked with *green asterisks*) remains localized in the nuclear fraction after oxidative stress. Immunoblots are representative of three independent experiments

changes in the cellular redox state and is, therefore, cysteine independent. Thus, nuclear TDP-43 oligomers are structurally distinct from the higher molecular weight species resulting from cysteine oxidation and disulfide bond formation, which are specifically formed in response to oxidative stress, predominantly in the cytosol[30, 32].

**Crystal structure of TDP-43 NTD shows superhelical packing**. Incorporation of TDP-43 in cytoplasmic stress granules is mediated via its C-terminal LCD[11, 22]. Moreover, the nuclear localization of TDP-43 oligomers, in conjunction with their resistance to oxidative stress (Fig. 1c–e) suggested that nuclear TDP-43 oligomerization is distinct and potentially structurally independent of

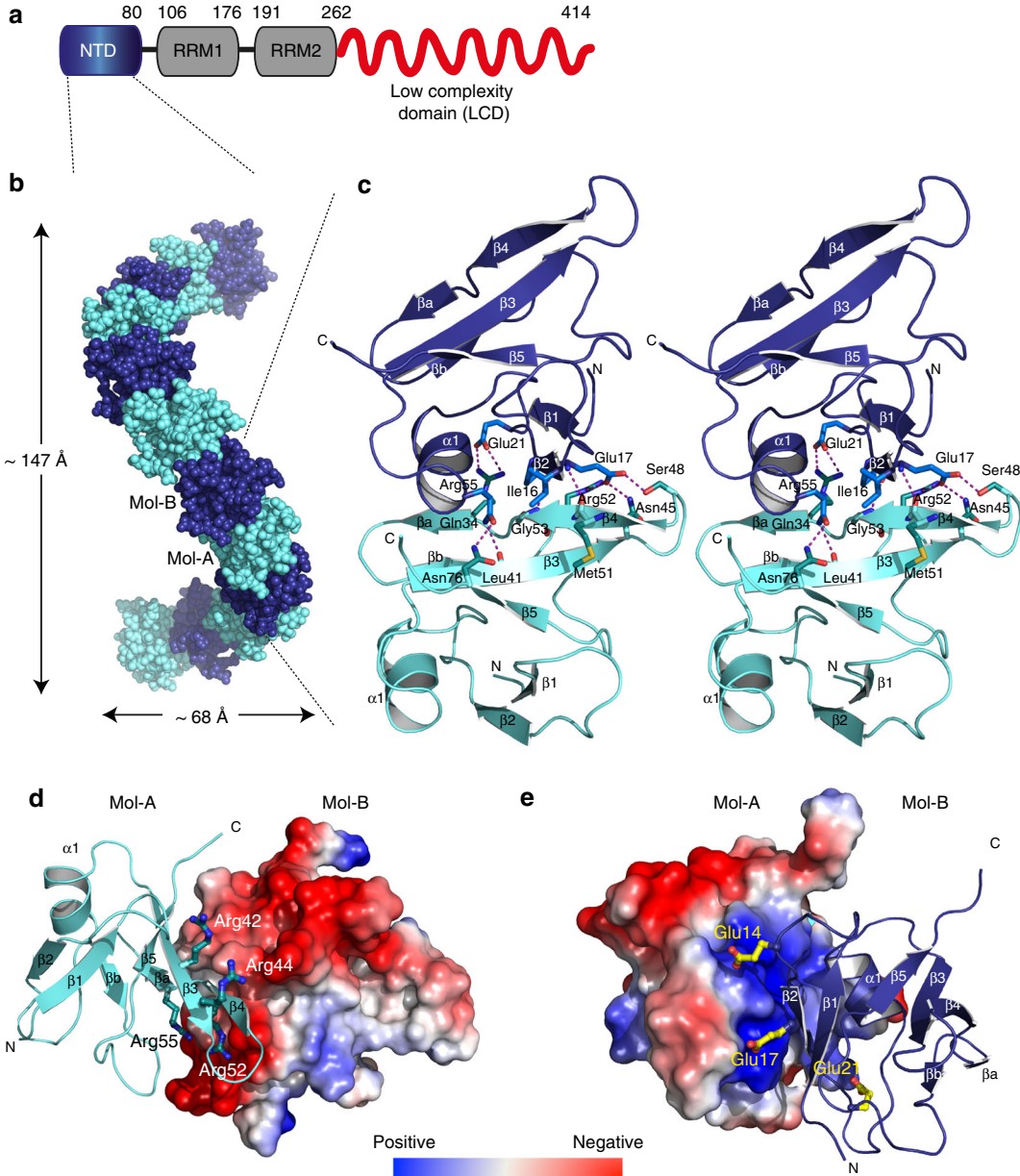

**Fig. 2** Crystal structure of TDP-43 NTD at 2.1 Å resolution. **a** Schematic domain organization of human TDP-43 with the N-terminal domain (*NTD*) shown in *blue*, RNA recognition motifs (*RRMs*) in *gray* and the low-complexity domain (*LCD*) in *red* with domain boundaries indicated on top. **b** Crystals of TDP-43 NTD show helical filaments with single molecules arranged in head-to-tail fashion as seen from the side. Atoms of Mol-A and -B (comprising asymmetric unit) are shown as *cyan* and *blue* spheres, respectively. The dimensions of the crystallographic helix are indicated. **c** Wall-eyed stereo view of two TDP-43 NTD molecules (Mol-A, Mol-B) in the asymmetric unit. The two molecules are shown in cartoon representation with the secondary structure elements and the N- and C-termini of the molecules labeled. The side chains making inter-molecular contacts are shown in stick representation in the corresponding domain color and labeled. Inter-molecular hydrogen bonds are shown as *dotted magenta lines*. Similar color and labeling schemes are used in other figures unless stated. **d**, **e** Electrostatic charge on the surface of two TDP-43 NTD molecules (Mol-A, Mol-B) of the asymmetric unit. Amino-acid residues in the positively charged head **d** and negatively charged tail region **e** are shown in stick representation and labeled. Positive potential is shown in *blue* and negative potential in *red*

LCD. We, therefore, focused on the NTD of TDP-43, whose role in protein dimerization was previously suggested[18, 19].

In order to understand the molecular basis of TDP-43 oligomerization, we aimed at determining a high-resolution structure of TDP-43 NTD. Based on the previously reported domain boundaries[15, 16], we expressed and purified recombinant human TDP-43 NTD (amino acids 1–80) in amounts amenable for structural analysis (Fig. 2a, Supplementary Fig. 3a). TDP-43 NTD was purified under native conditions, without the use of any denaturant during the purification process, in contrast to earlier

reports where some denaturants (such as urea) were included that may lead to destabilization of oligomers in solution[15, 16]. Subsequently, vapor diffusion crystallization experiments of purified recombinant protein were set up that led to initial crystallization hits. These small crystals were used for microseeding experiments to grow larger crystals (70 × 70 × 200 μm) that diffracted to 2.1 Å resolution (Supplementary Fig. 3b, Table 1). Initial molecular replacement trials using the deposited TDP-43 NTD NMR structure[16] (PDB ID code 2N4P) failed. However, crystals were grown in sodium cacodylate buffer and consequently

**Table 1 Data collection and refinement statistics**

| | TDP-43 NTD |
|---|---|
| *Data collection* | |
| Space group | $P6_3$ |
| Cell dimensions | |
| $a, b, c$ (Å) | 89.52, 89.52, 49.15 |
| $\alpha, \beta, \gamma$ (°) | 90, 90, 120 |
| Resolution (Å) | 44.76–2.10 (2.15–2.10) |
| $R_{meas}$ | 0.108 (2.03) |
| $R_{merge}$ | 0.103 (1.94) |
| $I/\sigma I$ | 16.7 (1.6) |
| $CC_{1/2}$ | 1.00 (0.59) |
| Completeness (%) | 98.8 (100.0) |
| Redundancy | 10.41 (10.92) |
| $\Delta_{ano}/\ \sigma\Delta_{ano}$ | 1.3 (0.7) |
| | |
| *Refinement* | |
| Resolution (Å) | 49.15–2.10 |
| No. of reflections | 12,500 |
| $R_{work}/R_{free}$ | 0.166/0.206 |
| No. of atoms | 1344 |
| Protein | 1192 |
| Ligand/ion | |
| Acetate | 1 |
| Cacodylate | 4 |
| Water | 108 |
| *B*-factors | |
| Protein | 48.46 |
| Ligand/ion | 54.16 |
| Water | 56.2 |
| R.m.s. deviations | |
| Bond lengths (Å) | 0.031 |
| Bond angles (°) | 2.72 |

One crystal was used for data collection. Values in parentheses are for the highest-resolution shell. Friedel pairs were treated as different reflections due to the significant anomalous signal.

starting phases were obtained by single-wavelength anomalous dispersion making use of the strong anomalous signal that was generated by the arsenic K-edge (Table 1). Using the anomalous signal, we obtained a substructure with four cacodylate ions that were covalently bound to surface exposed cysteine residues.

TDP-43 NTD crystallized in space group $P6_3$ and two molecules (Mol-A, Mol-B) are found in the asymmetric unit, whose structures are virtually identical, indicated by a root-mean square deviation (r.m.s.d) of 0.612 Å (Cα atoms of residues 1–80). Crystallographic symmetry generated three intertwined superhelices of TDP-43 NTDs packed against each other, building a tubular superstructure (Supplementary Fig. 3c, d). Each of these tubules is surrounded by another six tubules, thereby forming the lateral contacts of the crystal. Within each superhelix, every turn is composed of 12 TDP-43 NTD monomers (two molecules per asymmetric unit) with a rise of 24.6 Å between two asymmetric units, a helical pitch of 147.6 Å and a radius of 34 Å to the center of mass, respectively, (Fig. 2b).

**Head-to-tail interaction of TDP-43 NTD monomers**. Except for the N-terminal His-tag, all other residues were well resolved in the final electron density map. The structure of the TDP-43 NTD monomer is similar to the previously reported NMR structure with a global r.m.s.d of 1.97 Å for Mol-A and 2.12 Å for Mol-B for Cα atoms (Supplementary Fig. 3e), explaining the failure of solving the structure by molecular replacement. Significant differences are seen in the orientation and conformation of loops, especially the β3-β4 loop that seems to be flexible based on the NMR data[16], however, it is involved in inter-molecular interactions between monomers in

the crystal structure (Fig. 2c, Supplementary Fig. 3f). TDP-43 NTD adopts a compact fold comprised of a five-stranded β-sheet (β2-β1-β5-β3-β4) packing against one α-helix (Fig. 2c). An additional β-hairpin is present in the loop between β4- and β5-strands (βa and βb in Fig. 2c). Such domain architecture is reminiscent of Ubiquitin and DIX domains (see Discussion).

The inter-molecular interface between consecutive monomers is conferred by two distinct regions in the protein. The head region (on the right when viewing the β-sheet with the β1-strand pointing up) of TDP-43 NTD is positively charged and interactions are made by residues mainly from β3- (Asn45), β4- (Arg52, Arg55), β5- (Asn76) strands and the β3-β4 loop (Ser48) (Fig. 2c, d). In contrast, the tail (on the left, viewing from β-sheet with β1-strand going up) of the molecule is negatively charged (Fig. 2e) and is comprised of residues from the β2- (Glu17) strand, the α1-helix (Gln34) and the β1-β2 (Glu14) and β2-α1 (Glu21) loops that interact with the adjacent molecule (Fig. 2c, e). Such interaction between NTD molecules results in charge complementation. In addition, at each interface, β2-strand of one monomer forms a parallel inter-molecular β-bridge with the β4-strand of the adjacent monomer in which the main chain atoms of Glu17 (β2-strand) forms H-bonds with the backbone of Met51 and Gly53 (β4-strand) (Fig. 2c, Supplementary Fig. 4a, b). This interaction is strengthened by the guanidinium groups of Arg55 and Arg52 that form salt bridges with the carboxyl groups of Glu21 and Glu17, respectively (Fig. 2c, Supplementary Figs. 4b and 5a, b). Additionally, the carboxyl group of Glu17 forms specific hydrogen bonds with the hydroxyl group of Ser48 and the carboxamide group of Asn45 (Fig. 2c, Supplementary Figs. 4b and 5a). Moreover, the aliphatic side chains of Ile16 and Pro19 (in Mol-B) interact with the aliphatic parts of side chains of Met51, Arg52, and Arg55 (in Mol-A), respectively, to create a small hydrophobic contact at the interface (Fig. 2c, Supplementary Fig. 4b). Finally, the α1-helix of one monomer packs against the β-sheet of the other molecule where in the carboxamide of Gln34 (α1-helix) makes hydrogen bonds to the carboxamide nitrogen of Asn76 (β5-strand) and the backbone carbonyl oxygen of Leu41 (β3-strand) (Fig. 2c, Supplementary Figs. 4a and 5c).

Thus, the two NTD molecules in the asymmetric unit interact extensively with one another in head-to-tail fashion (Fig. 2c, Supplementary Fig. 3f), burying a total surface area of 644.2 Å² at the interface between Mol-A and Mol-B within the asymmetric unit and 648.1 Å² between Mol-B and Mol-A' between two adjacent asymmetric units (the prime refers to a molecule related by crystallographic symmetry). This area is comparable to the buried surface area in the oligomeric structures formed by the DIX domains of both Axin-1 (768 Å²) and Dishevelled Segment Polarity Protein-2 (425–470 Å²).

**Dynamic polymerization of TDP-43 NTD revealed by NMR**. In order to confirm and validate the interactions observed in the crystal structure, we utilized NMR spectroscopy. At 50 μM protein concentration, isotopically labeled TDP-43 NTD appears to be clearly folded in solution as indicated by the wide NMR chemical shift dispersion in the two-dimensional (2D) $^1$H-$^{15}$N Heteronuclear Single Quantum Coherence (HSQC) spectrum (Fig. 3a). Interestingly, upon increasing the protein concentration, a subset of cross peaks of TDP-43 NTD undergoes chemical shift perturbations (CSPs) and increased spectral line widths (Fig. 3a). Sequence-specific resonance assignments of TDP-43 NTD (Supplementary Fig. 5d) revealed that these perturbations are mainly corresponding to residues I18 (in β2-strand), Val31, Thr32, Ala33, and Gln34 (in α1-helix), Cys39, Gly40 and Asn45 (in β3-strand), Arg52 and Gly53 (in β4-strand) and Val75 and

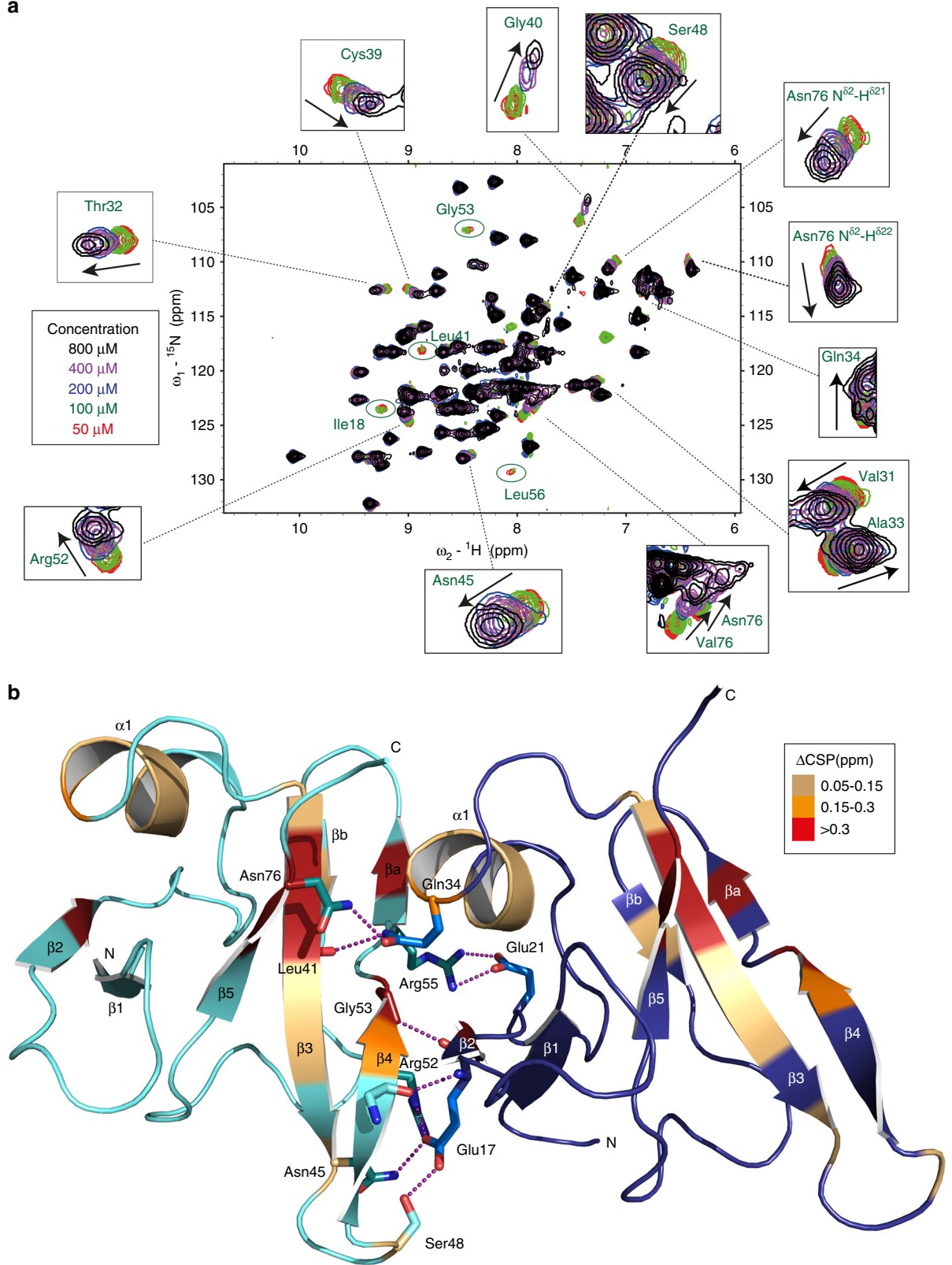

**Fig. 3** TDP-43 NTD oligomerization analyzed by solution NMR spectroscopy. **a** Overlay of 2D $^1$H-$^{15}$N HSQC spectra at 30 °C of $^{15}$N isotopically labeled TDP-43 NTD at protein concentrations of 50, 100, 200, 400, and 800 μM in *red, green, blue, magenta,* and *black,* respectively. Magnified views of cross peaks undergoing chemical shift perturbations (CSP) are shown around the full spectra overlay while peaks showing line broadening are encircled in *green* and labeled. *Black arrows* show the direction of movement of the peaks. **b** Backbones of amino acids undergoing CSP are colored on the cartoon of TDP-43 NTD crystal structure. Relative to the lowest protein concentration (50 μM), the changes in average chemical shift perturbations ($\Delta$CSP = $\Delta(((\delta$H$)^2 + (\delta$N$/5)^2)/2)^{1/2}$, in which δH and δN are in ppm) were calculated at highest protein concentration (800 μM) and colored on the structure based on the magnitude of perturbations as indicated. Side chains contributing to inter-molecular interface seen in the crystal structure are displayed as sticks with H-bonds in *magenta*

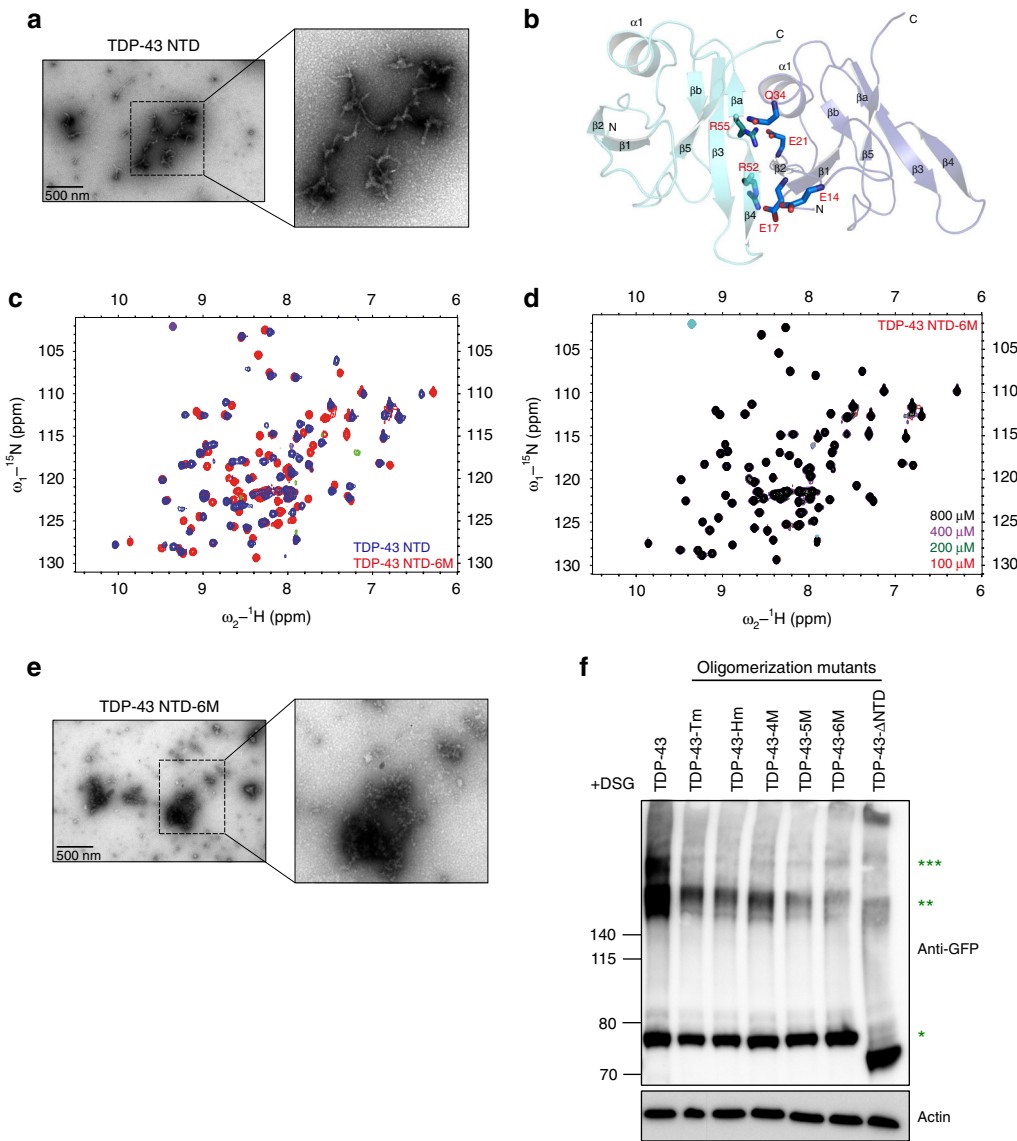

**Fig. 4** Point mutations at oligomerization interface abrogate TDP-43 polymerization in vitro and in cells. **a** Transmission electron microscopy (TEM) image of TDP-43 NTD fibrillar oligomers following negative staining. One region (*dotted black box*) from the image is shown magnified on the *right*. **b** Position of the six amino acids that were substituted to abolish oligomerization are shown on the cartoon of TDP-43 NTD crystal structure with side chains shown as sticks and labeled in *red*. **c** Overlay of 2D $^1$H-$^{15}$N HSQC spectra of wild type TDP-43 NTD (*blue*) and oligomerization mutant TDP-43 NTD-6M (*red*) at protein concentration of 100 μM at 30 °C. **d** Overlay of 2D $^1$H-$^{15}$N HSQC spectra of TDP-43 oligomerization mutant TDP-43 NTD-6M at increasing protein concentration of 100, 200, 400, and 800 μM in *red, green, magenta* and *black*, respectively at 30 °C. **e** TEM image showing the lack of fibrillar structures for the oligomerization mutant TDP-43 NTD-6M. One region (*dotted black box*) from the image is shown magnified on the *right*. **f** Immunoblots (from 4–12% denaturing polyacrylamide gel) with anti-GFP antibody (*upper panel*) of lysates obtained from cells transiently expressing wild-type GFP-tagged human TDP-43 (GFP-TDP-43) or oligomerization mutants (GFP-TDP-43-Tm/Hm/4 M/5 M/6 M/ΔNTD) followed by DSG-mediated cross-linking. Oligomerization mutants display significantly reduced high-molecular weight oligomeric TDP-43 compared to wild-type TDP-43. Actin is used as protein loading control (*lower panel*). Immunoblots are representative of three independent experiments

Asn76 (in β5-strand, Fig. 3a). These are fully consistent with the inter-molecular interface seen in the crystal structure (Fig. 3b). Furthermore, Gly53, which is involved in an inter-molecular β-bridge, shows significant line broadening that goes beyond detection at higher protein concentrations (green circle in Fig. 3a). Interestingly, this process is reversible because upon dilution the cross-peaks become narrower and shift back to their original position showing that this oligomerization of TDP-43 NTD is dynamic in solution. The CSPs are in fast to intermediate exchange regime on the NMR time scale and the concentration dependence range implies that the affinity of NTD monomer self-association is in the low micromolar range, which

is comparable to the reported dissociation constant for DIX, Phox and Bem1 (PB1) domains (5–20 μM for Axin-1, 200 μM for Axin-1-Dvl-2 and 100 μM for Par-3-PB1 domain)[33–35].

**Visualization of polymeric TDP-43 NTD by electron microscopy.** To directly visualize the TDP-43 NTD polymers under native conditions, we used transmission electron microscopy (TEM) on negatively stained samples of purified recombinant protein in the buffer conditions used for NMR experiments. TDP-43 NTD formed fibrillar structures of variable length and diameter, even at low protein concentrations of 100 μM (Fig. 4a). Some filaments are about 7 nm in diameter, which corresponds to the width of

the helical filament seen in the crystal structure, while the observed thicker filaments may result from coalescence of individual filaments. Taken together, the head-to-tail interaction interface seen in the crystal structure and confirmed in solution by NMR and the ability of TDP-43 NTD to form fibrillar structures in native buffer may explain the intrinsic

oligomerization property of TDP-43 seen in vivo (Fig. 1a, b). The negatively charged tail (Glu14, Glu17, and Glu21) and the positively charged head (Arg52 and Arg55) interaction interface are evolutionary conserved (Supplementary Fig. 6a), suggesting a conserved mode of TDP-43 oligomerization across various species.

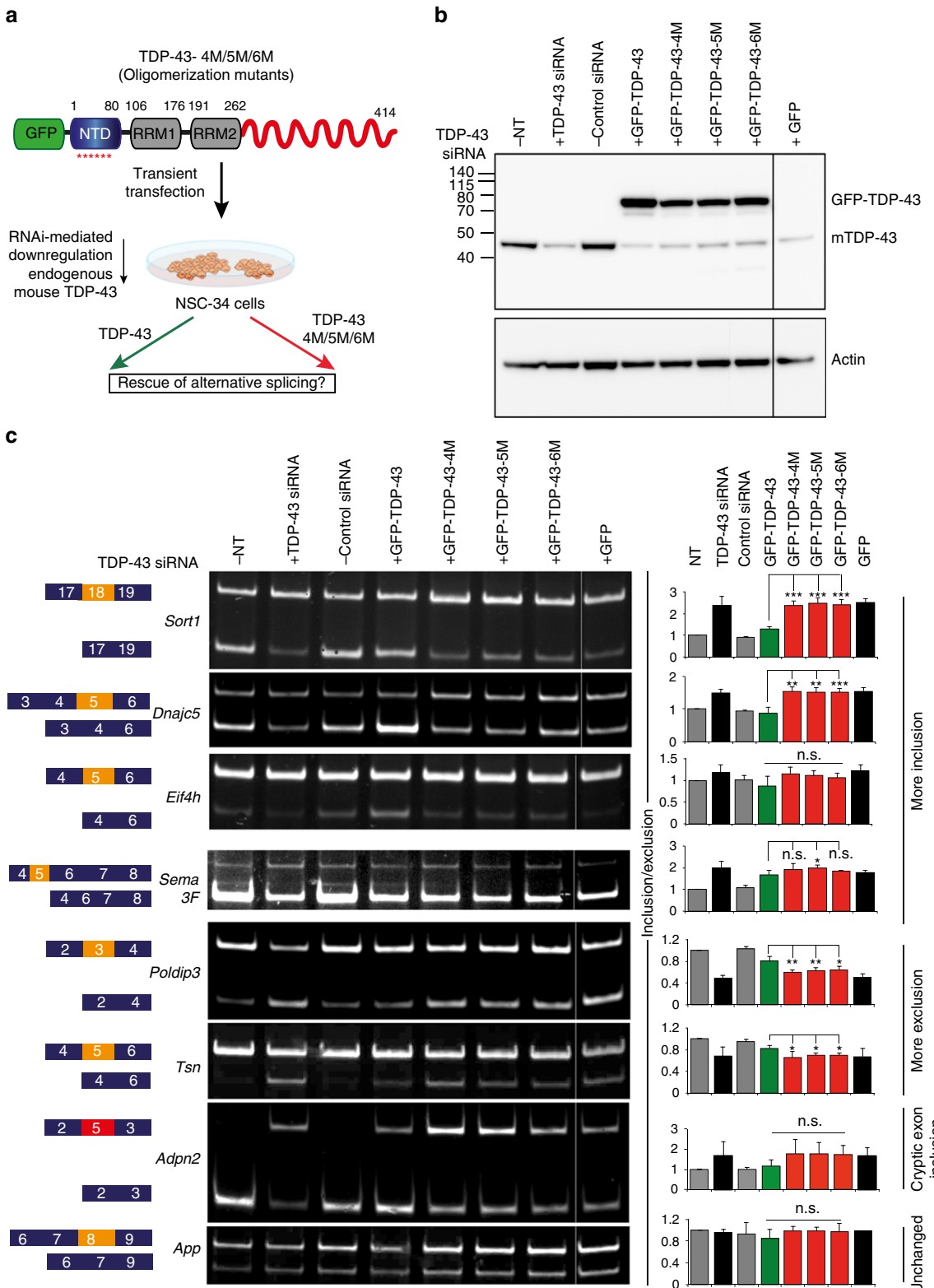

**Interface disrupting mutations abolish TDP-43 polymerization.**
In order to investigate the functional relevance of the interaction interface observed in the crystal structure and in solution, we designed, expressed and purified recombinant proteins with specific point mutations that disrupt the oligomerization interface (Fig. 4b, Supplementary Fig. 6b–c), without affecting the folded state of the protein (Fig. 4c, Supplementary Fig. 6d). Based on the crystal structure, interference with one charged surface should be sufficient to prevent protein oligomerization in vitro. Indeed, both the double mutant E17A/E21A in the tail region (Tm) and R52A/R55A in the head region (Hm) and different combinations of head and tail mutants (designated 4, 5, and 6 M; Supplementary Fig. 6b) abolish oligomerization of the protein in vitro, as monitored by NMR spectroscopy and TEM (Fig. 4d, e, Supplementary Fig. 7a–d). Significant changes in the 2D $^1$H-$^{15}$NHSQC spectra of mutant proteins compared to wild-type NTD occurred, due to disruption of the interaction interface (Fig. 4c, Supplementary Fig. 6d). Furthermore, the narrow and uniform spectral line widths of mutant proteins compared to the wild type protein, remaining at even higher protein concentrations (800 μM) (Fig. 4d) strongly suggested their monomeric nature in solution. Most importantly, no changes in chemical shifts were observed upon increasing protein concentration of isotopically labeled mutant proteins (Fig. 4d, middle panel in Supplementary Fig. 7a–d). Additionally, as analyzed by electron microscopy, none of the mutant TDP-43 NTD formed any fibrils but predominantly formed amorphous deposits in the supporting matrix used in the TEM studies (Fig. 4e, right panel in Supplementary Fig. 7a–d).

To investigate the role of NTD-driven TDP-43 oligomerization in cells, we introduced interface-disrupting point mutations in the context of GFP-tagged full-length human TDP-43 in mouse motor neuron-like NSC-34 cells. To prevent oligomerization of overexpressed protein with endogenous mouse TDP-43, which is 96% identical to human TDP-43, we used point mutations in both head and the tail region (mutants designated 4, 5, and 6 M; Fig. 4b, Supplementary Fig. 6b). Oligomerization of transiently expressed protein was analyzed following DSG cross-linking in cells. The mutations severely abrogate the ability of overexpressed protein to form higher molecular weight TDP-43 oligomers with most severe disruption of oligomerization observed in TDP-43-6M, where both head and tail regions are mutated simultaneously (Fig. 4f, Supplementary Fig. 7e). These data suggest that cellular, full-length TDP-43 oligomerizes via its NTD utilizing the same protein–protein interface identified in the crystal structure and solution studies.

**TDP-43 polymerization is crucial for RNA splicing regulation.**
TDP-43 has been shown to regulate splicing of specific RNA targets by binding to proximal intron-exon junctions of pre-mRNA[6, 36, 37]. Indeed, TDP-43 downregulation in mouse brain resulted in characteristic splicing alterations of target mRNAs[6]. To validate the inter-molecular interactions seen in the crystal structure and assess their functional importance, we analyzed the ability of oligomerization mutants to rescue the splicing defects induced by endogenous TDP-43 depletion in mouse NSC-34 cells (Fig. 5a). We employed a previously reported TDP-43-specific siRNA[38], which reduced endogenous mouse TDP-43 levels by > 90%, without interfering with expression of co-transfected RNAi-resistant wild-type or oligomerization-deficient human TDP-43 mutants (Fig. 5b). RNAi resistance was accomplished by scrambling the TDP-43 target sequence of siRNA, without altering the coded protein sequence. Transiently expressed wild-type and oligomerization-deficient TDP-43 proteins showed similar expression levels and cellular localization both in the nucleus and the cytoplasm of transfected cells (Supplementary Fig. 8a–g), allowing a comparison in their ability to rescue the RNA splicing function in the nucleus. Consistent with our previous findings[6], downregulation of TDP-43 resulted in missplicing of specific RNA targets, such as Sort1 (Sortilin1), Dnajc5 (DnaJ homolog subfamily C member 5), Poldip3 (polymerase delta-interacting protein 3), Eif4h (eukaryotic translation initiation factor 4H), Sema3 (semaphorin-3F), and Tsn (translin), which was rescued by co-transfection with wild type TDP-43 (Fig. 5c). In contrast, the oligomerization-deficient mutants failed to rescue RNA missplicing, strongly indicating that head-to-tail NTD-driven TDP-43 oligomerization is essential for its function in RNA splicing. We observed a similar pattern in rescuing splicing defect in Adnp2 that has been reported to express a cryptic exon upon TDP-43 downregulation[39] (Fig. 5c). As a negative control, we assessed the splicing pattern of App, which is a direct RNA target of TDP-43, but whose splicing is not regulated by TDP-43[6]. Additionally, point mutations away from the oligomerization interface could rescue the splicing defect similar to wild type TDP-43 (Supplementary Fig. 9a–c). Furthermore, consistent with in vitro experiments (Fig. 4f, Supplementary Fig. 7a, b), mutation of only two residues (one charged interface) is sufficient to abolish the function of TDP-43 in splicing, highlighting their role in the oligomerization-dependent function of the protein (Supplementary Fig. 9d, Supplementary Fig. 10a–h).

**Dynamic TDP-43 polymerization impedes pathologic aggregation.**
Based on our structural and functional data, we hypothesized that the NTD-mediated, dynamic, physiological oligomerization might orient the LCDs of two adjoining TDP-43 molecules within the oligomer apart from each other. Such orientation would prevent inter-molecular LCD interactions and might potentially antagonize irreversible protein aggregation and phosphorylation. To experimentally test this and determine the relative spatial location of N- and C-terminal regions of adjacent TDP-43 molecules within oligomers, we used the tripartite split GFP complementation system[40] in mouse immortalized multipotent neural progenitor C17.2 cells (Fig. 6a). For this, we tagged either the N- or the C-termini of

**Fig. 5** TDP-43 oligomerization is essential for its role in RNA metabolism. **a** Schematic overview of rescue experiment to assess the role of TDP-43 oligomerization in RNA splicing. Point mutations interfering with TDP-43 oligomerization are schematically marked by *red asterisks* below the NTD. **b** Immunoblot (from 12% denaturing polyacrylamide gel) using an antibody against mouse/human TDP-43, showing specific downregulation of endogenous mouse TDP-43 and overexpression of RNAi resistant GFP-tagged wild type and oligomerization mutant human TDP-43 (GFP-TDP-43-4M/5 M/6 M) in mouse NSC-34 cells (*upper panel*). Actin is used as protein loading control (*lower panel*). Immunoblots are representative of four independent experiments. **c** Semi-quantitative RT-PCR analysis of selected target's alternative splicing regulated by TDP-43. *Left panel* depicts alternatively spliced exons (*orange*) or cryptic exons (*red*) flanked by their constitutive exons in *blue boxes*. *Middle panel* shows representative polyacrylamide gel images of semi-quantitative RT-PCR products for all conditions. *Right panel* shows quantification of splicing changes plotted as the ratio of exon inclusion/exclusion (on *y*-axis) against different conditions (on *x*-axis) averaged from four biological replicates and normalized to the ratio obtained for not treated (NT) cells that is arbitrarily set to 1. In all cases, the expression of wild-type TDP-43 (*green bars*) rescues the changes in splicing caused by downregulation of endogenous mouse TDP-43 (TDP-43 si-RNA, *black bars*). The effect of each oligomerization mutant TDP-43 is shown (*red bars*). Statistical comparison of the inclusion/exclusion ratio between wild type and oligomerization mutants (4 M, 5 M, or 6 M) is indicated by *asterisks* above respective bars (*$p < 0.05$, **$p < 0.01$, ***$p < 0.001$, two-tailed Student *t*-test). Standard deviation is represented by error bars. Full gels with molecular weight markers are shown in Supplementary Fig. 10

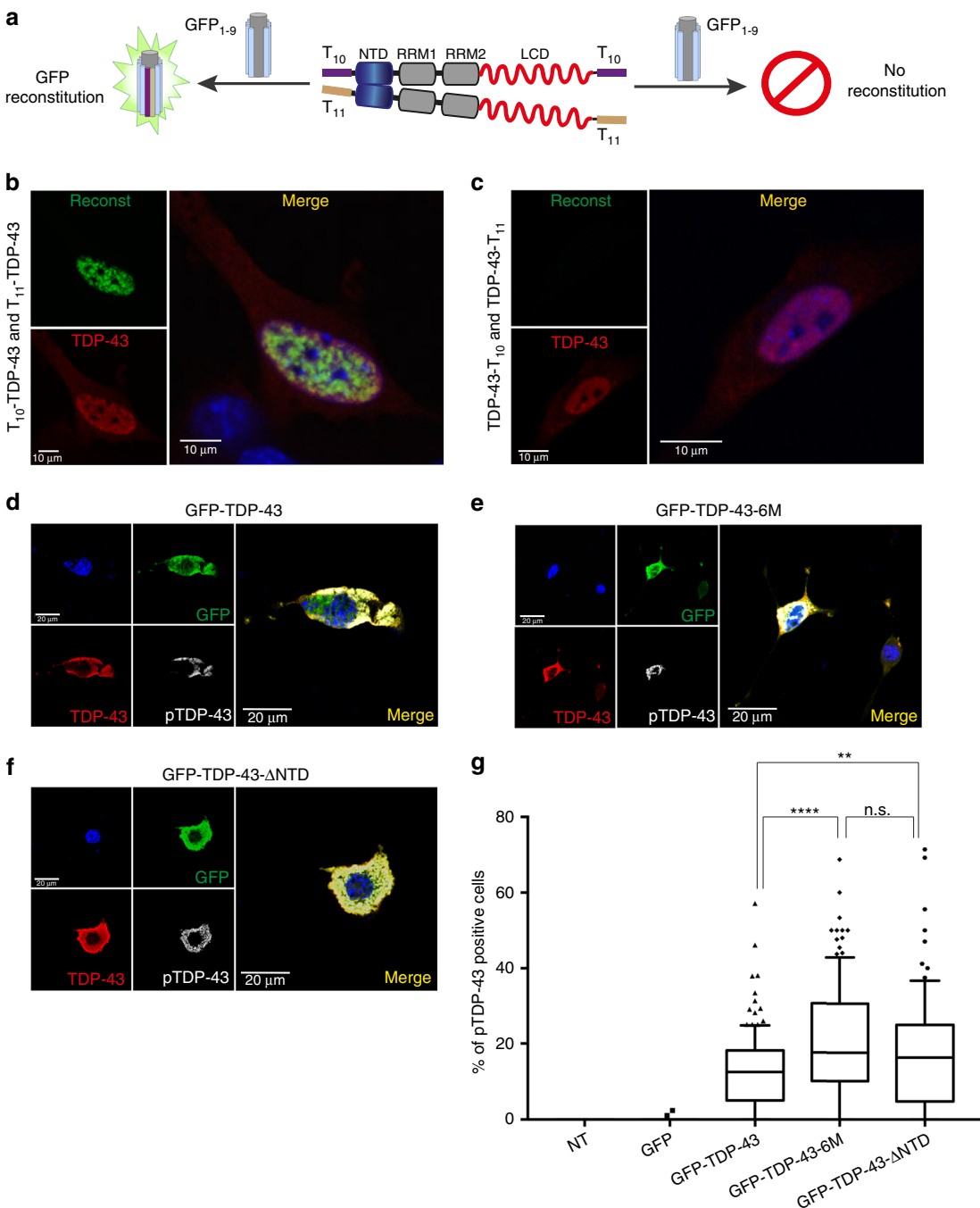

**Fig. 6** NTD-mediated TDP-43 oligomerization impedes inter-molecular LCD interactions and antagonizes pathologic protein aggregation. **a** Schematic of tripartite GFP complementation assay. TDP-43 was tagged either at the N- or C-terminus with the 10th ($T_{10}$) or 11th ($T_{11}$) β-strand of GFP (schematically indicated in *magenta* and *light yellow*, respectively). Successful complementation results in fluorescence reconstitution with $GFP_{1-9}$ molecule (GFP β-strands 1–9 shown in *blue* and α-helix in *grey*). **b**, **c** N- **b** or C-terminally **c** tagged $T_{10}$- and $T_{11}$- TDP-43 were co-transfected with $GFP_{1-9}$ in mouse C17.2 cells. Confocal microscopy analysis on cells shows successful complementation as fluorescence reconstitution in case of N-terminally tagged $T_{10}$- and $T_{11}$- TDP-43 **b**. In contrast, the C-terminal tagged $T_{10}$- and $T_{11}$- TDP-43 fail to reconstitute fluorescence **c**. Merged and zoomed in images are shown on the *right*. Transfected mouse C17.2 cells were counter-stained with a human-specific TDP-43 antibody (*red*) and the nuclear marker DAPI (*blue*).
**d**–**f** Confocal microscope images of NSC-34 cells overexpressing GFP-tagged wild type **d** or oligomerization defective TDP-43 mutants **e**, **f**. Cells were stained with anti-GFP (*green*), anti-TDP-43 (*red*) and anti-phosphorylated TDP-43 (*gray*) antibodies. Merged images overlaid with DAPI (*blue*) are shown on the *right*. **g** The percentage of phosphorylated TDP-43 (pTDP-43)-positive cells is plotted for wild-type and oligomerization mutant TDP-43 (TDP-43-6M and TDP-43-ΔNTD). Manual cell counting was conducted on confocal images (20× magnification). For each condition, approximately 2500 GFP-positive cells (or 1500 cells for GFP-TDP-43 ΔNTD) from three biological replicates were counted by an investigator that was blinded to the identity of the samples. Percentage of phosphorylation was determined considering the total number of GFP-positive transfected cells. An unpaired student *t*-test was performed to determine significance (depicted as *asterisks*). *P*-values: WT vs. 6 M: <0.0001 (****), WT vs. ΔNTD: 0.003 (**), 6 M vs. ΔNTD: 0.184 (n.s)

TDP-43 with the 10th ($T_{10}$) or the 11th ($T_{11}$) β-strand of GFP, which when in close proximity to each other can reconstitute fluorescence of the remaining $GFP_{1-9}$ protein, which by itself lacks fluorescence ability[40] (Fig. 6a). We observed fluorescence reconstitution with $GFP_{1-9}$ only when $T_{10}/T_{11}$ were both positioned at the N-terminus of TDP-43 ($T_{10}/T_{11}$-TDP-43, Fig. 6b), but not at the C-

terminus (TDP-43-$T_{10}/T_{11}$, Fig. 6c), or any other N- or C-terminal combinations (Supplementary Fig. 11a). These data indicate that indeed the C-termini of the adjoining molecules within TDP-43 oligomers are distantly located from each other, as opposed to the N-termini that comprise the oligomerization interface within cells. To rule out the loss of fluorescence reconstitution due to

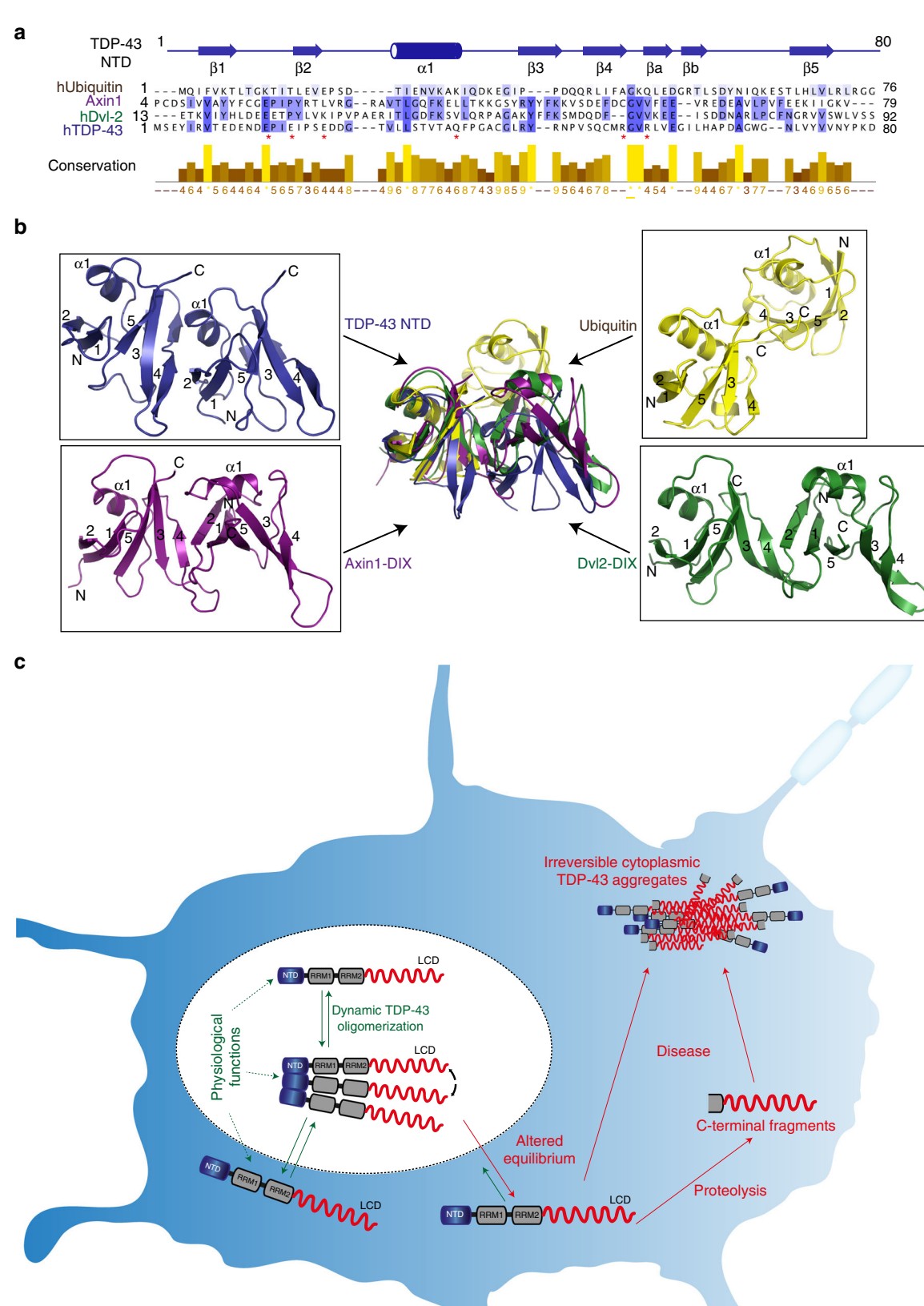

inaccessibility of the TDP-43 C-terminal region, we co-transfected either the N- or C-terminal $T_{10}/T_{11}$ TDP-43 with $GFP_{1-10}$, which confirmed complementation of $T_{11}$-tagged TDP-43, independently of its fusion site (Supplementary Fig. 11b–d). This suggests, that a lack of fluorescence reconstitution with $GFP_{1-9}$ at C-terminally tagged $T_{10}/T_{11}$ is indeed due to distant spatial organization of the TDP-43 C-termini in the cellular oligomers.

Collectively, the above data confirms that nuclear and functional TDP-43 oligomerization is formed via the interaction of its NTD and that within these oligomers the C-terminal LCDs of adjoining molecules are spatially separated. Since LCD interactions are necessary for cytoplasmic TDP-43 accumulation[14, 29], we reasoned that NTD-mediated oligomerization may antagonize its aggregation. To test this hypothesis, we compared the ability of wild-type and oligomerization-deficient TDP-43 to be recruited into cytoplasmic phosphorylated TDP-43 inclusions. Following transient overexpression of human GFP-tagged TDP-43, a small fraction of NSC-34 cells develop cytoplasmic aggregates that become abnormally phosphorylated, reminiscent to the characteristic pathology found in sporadic ALS patient tissue[2]. While image analysis of over 2500 control cells—either non-transfected, or GFP-transfected—failed to detect any phosphorylated TDP-43 signal, ~ 15–20% of wild type TDP-43-expressing cells showed phosphorylated TDP-43 inclusions, potentially driven by overexpression and increased cytoplasmic concentration of the nucleo-cytoplasmic shuttling protein (Fig. 6d, Supplementary Fig. 12). Remarkably, the percentage of cells showing cytoplasmic phospho-TDP-43 aggregates is significantly and consistently higher in both oligomerization-defective mutants (TDP-43-6M and TDP-43-ΔNTD), compared to wild type (Figs. 6e–g), suggesting an increased propensity of oligomerization mutants to form cytoplasmic aggregates. To test for increased aggregation in oligomerization mutants, we performed sequential insolubility assay (SIA) exposing the cell lysates to detergents of increasing strength (Supplementary Fig. 13a). Consistent with increased pTDP-43 in oligomerization mutants, the mutant protein also shows enhanced insolubility compared to the wild type, reflecting its enhanced tendency to aggregate (Supplementary Fig. 13b, c).

In conclusion, our high-resolution structural, biochemical, and cellular analyses uncover a novel NTD-mediated dynamic and functional TDP-43 oligomerization, which antagonizes its irreversible pathologic cytoplasmic aggregation.

## Discussion

TDP-43 is a multifunctional and essential RBP, whose abnormal cytoplasmic misaccumulation is integrally linked to the pathogenesis of ALS and FTD[1, 2]. In this study, we provide evidence that the physiological TDP-43 state in the nucleus is oligomeric and that this oligomerization is indispensable for the functional role of the protein in RNA metabolism. Dimerization or oligomerization of RBPs can occur in an RNA-dependent or independent fashion[41]. Our crystal structure shows that NTD-driven TDP-43 oligomerization is independent from RNA binding, although the potential increase in local TDP-43 concentration upon RNA binding within cells may well favor oligomerization. Nevertheless, oligomeric TDP-43 would be expected to increase both its affinity and specificity for its RNA targets, compared to a monomeric form. This is in agreement with the observation from our previous CLIP-seq data in mouse brain[6], showing that in most TDP-43 RNA targets, the UG-repeats are longer than required for binding of TDP-43 monomer, whose "footprint" on RNA comprises of 3–4 UG repeats[6]. In fact, 63% of total TDP-43-binding sites identified by CLIP-seq occur within long intronic regions and this binding sustains the levels of RNAs important for neuronal function[6]. An attractive hypothesis that emerges from our current study is that TDP-43 binds these regions preferentially as an oligomer to increase affinity towards long contiguous UG-repeats. Direct RNA binding of the NTD may contribute to this, as suggested by the low affinity binding of this domain to TG repeats by gel shift assay (Supplementary Fig. 14a), consistent with previous reports[18] showing some—albeit low—contribution of TDP-43 NTD to RNA binding. Moreover, TDP-43 oligomerization may act as a "recruitment platform" for specific factors involved in splicing and RNA maturation steps important for sustaining their levels. Another potential role of NTD-driven inter-molecular TDP-43 interaction is the ability to bring distal sites on RNA in close proximity, resulting in looping-out of RNA. Depending on the binding sites, this mechanism may promote inclusion or skipping of alternative exons. Such a mechanism has previously been proposed for other RBPs such as PTB[42], hnRNP A/B, hnRNP F/H[43], and STAR proteins (T-star and Sam68)[44].

TDP-43 oligomerization is novel and unique among the higher order structures described for other RBPs and is mediated by NTD, whose structural homologs have not been reported in any other RBP until now. However, a sub-set of protein–protein interaction domains such as DIX, PB1, and sterile alpha motif (SAM) domains undergo similar head-to-tail interactions to generate not only homo- but also heteropolymers that have been implicated in modulating signalosome assembly[24]. Interestingly, despite very low sequence homology (Fig. 7a), TDP-43 NTD displays high structural similarity with both the monomer structure (Supplementary Fig. 14b), and the polymer structure of DIX domains (Fig. 7b, Supplementary Fig. 14c–e), while only resembling the monomeric structure of ubiquitin as proposed earlier[15]. However, while the interaction interface is comprised of similarly positioned structural elements in DIX domains, the molecular interactions are different in TDP-43 NTD (Fig. 7a). Nevertheless, based on high structural resemblance to DIX domains, it is conceivable that TDP-43 NTD might as well modulate specific signaling events by regulating the high local concentration of the protein. Interestingly, TDP-43 has been shown to interact with the signaling molecule nuclear factor (NF)-κB subunit p65 via its NTD[38] and it would be important to clarify if this interaction depends on TDP-43 oligomerization.

**Fig. 7** Novel TDP-43 oligomerization domain and implications for disease. **a** Multiple sequence alignment of TDP-43 NTD with structurally homologous DIX domains (Axin-1, Dvl-2) and Ubiquitin. Alignment was performed by T-coffee (http://www.ebi.ac.uk/Tools/msa/tcoffee/) and the alignment figure was generated in Jalview (http://www.jalview.org/). TDP-43 NTD secondary structural elements are depicted above the sequence and residues involved in the inter-molecular interactions within TDP-43 NTD are marked by *red asterisks* below the sequence. Sequence conservation scores are shown below the alignment as color gradient (*dark yellow*—poor conservation, *light yellow*—high conservation). **b** Overlay of structures of TDP-43 NTD (in *blue*) with Axin-1 DIX (in *magenta*, PDB ID 1WSP), Dvl2-DIX (in *green*, PDB ID 4WIP), and Ubiquitin (in *yellow*, PDB ID 1TBE). Structures are overlaid on one molecule to reveal the differences in the orientation of the second molecule. Individual structures in the same orientation are shown around the overlay. **c** Role of dynamic nuclear TDP-43 polymerization in physiological conditions and in disease. Within the nucleus, TDP-43 oligomerizes via the NTD and such conformation prevents irreversible aggregation by spatially separating the LCDs of adjoining TDP-43 molecules. Disturbance in the equilibrium between the oligomeric and monomeric TDP-43 in the cytoplasm (shown by *green-red arrows*) may result in proteolytic cleavage of exposed monomeric TDP-43 and generation of C-terminal fragments lacking the NTD. These highly aggregative fragments may initiate formation of irreversible pathologic inclusions via the LCD

The mechanism of transition of physiological nuclear TDP-43 to pathologic, irreversible and insoluble protein assemblies remains enigmatic, but it is likely triggered by the proteolytic release of its C-terminal LCD, which harbors most of the ALS-linked mutations[1] and possesses high propensity to self-associate[11, 29], phase separate[12, 13], and aggregate[2]. Our current data strongly support the hypothesis that formation of physiological NTD-mediated, nuclear TDP-43 oligomers can counteract cytoplasmic aggregation (Fig. 7c). Any factor altering the cellular equilibrium between oligomeric and monomeric TDP-43 may trigger the initiation of cytoplasmic TDP-43 aggregation (Fig. 7c). It is conceivable that the NTD-driven oligomeric state protects TDP-43 from proteolytic cleavage, a process characteristic for pathologic TDP-43 in ALS patients[2]. Such cleavage, releases TDP-43 LCD, which no longer possesses the ability to form dynamic oligomers due to lack of the essential NTD. Therefore, TDP-43 LCD can act as a potent "seed" to trigger the formation of pathologic TDP-43 aggregates, which are comprised of both full-length and C-terminal TDP-43 fragments in ALS patients[2] (Fig. 7c).

Finally, although the relative contribution of loss of normal nuclear TDP-43 function and gain of toxicity of its cytoplasmic aggregates remains unknown, a combinatorial mechanism may well operate in disease, which precludes many possibilities for interventions that target either loss or gain of function exclusively. We propose that stabilizing TDP-43 functional oligomers could be an attractive therapeutic strategy to counteract irreversible cytoplasmic TDP-43 aggregation, while preserving its normal nuclear function.

## Methods

**Human samples.** Frozen autopsy samples of non-neurological control human cerebellum and motor cortex were provided by Dr John Ravits from University of California, San Diego and all patient consent was taken in the clinics.

**Cross-linking in cells and tissues.** Cross-linking in cells and tissues was performed by using DSG[25], which is a membrane permeable molecule that can passively diffuse inside living cells. DSG stabilizes native protein–protein interactions by forming irreversible covalent bonds between primary amine groups in the proteins. For cross-linking in cells, mouse NSC-34 and human fibroblasts were first harvested using trypsin (Gibco) and its subsequent inhibition by Trypsin inhibitor (Gibco). The cells were washed twice with phosphate-buffered saline (PBS, pH 7.4) and then resuspended in 100 μL PBS. Fresh 20 mM DSG (Sigma) stocks were prepared in dimethylsulphoxide. The reaction was started by adding an appropriate volume of the DSG stock to the 100 μL cell suspension (with Roche protease inhibitor cocktail in PBS) at a final DSG concentration of 50, 100, 250, 500, and 1000 μM. The reaction was incubated at room temperature with shaking for 30 min and then quenched by addition of Tris base (at a final concentration of 20 mM) for 15 min at room temperature (RT).

For cross-linking in human and mouse brain (frozen, post-mortem human cerebellum, motor cortex, and mouse organotypic slices), tissue samples were chopped twice on a Tissue Chopper (McIlwain Tissue Chopper, setting blade distance to 100 μm). In between each chopping, the sample was rotated by 90° to ensure fine mincing of the brain tissue. The finely minced brain samples were then transferred to 15 mL polypropylene tubes (GreinerBio-One) and resuspended in 5 mL PBS/PI by gentle pipetting (3–4 times). From this tissue suspension, homogenous aliquots were transferred to 1.5 mL eppendorf tubes (Protein LoBind, Eppendorf), that were centrifuged at 1500xg for 5 min at RT. Supernatants were rejected, and the pellets underwent cross-linking by addition of 1000 μL of 1 mM DSG in PBS/PI for 100 mg tissue. The samples were incubated at 37 °C for 30 min with shaking followed by quenching with 20 mM Tris base (final concentration) for 15 min.

**Nucleo-cytoplasmic fractionation.** Intact cell pellet or human and mouse tissue pellet following cross-linking were lysed in lysis buffer (50 mM Tris pH 7.4, 10 mM NaCl, 0.5% NP-40, 0.25% TritonX-100, 1 mM EDTA, protease inhibitors (Roche)) by incubating 5 min on ice[37]. After centrifugation at 3000xg for 5 min at 4 °C, the supernatant was collected as the cytoplasmic fraction and the pellet as the nuclear fraction.

**Oxidative stress treatment on mouse brain slices.** Three-hundred fifty micrometer thick brain slices were prepared from 5–8 day old C57BL/6J pups according to previously published protocols modified for cortico-hippocampal slices[45]. Two slices each were cultured in Millicell-CM inserts (Millipore) in forebrain culture medium (50% BME, 25% Earle's salt solution, 25% heat inactivated horse serum, supplemented with 1% glucose, penicillin/streptomycin,

and Glutamax (Invitrogen)). Slice cultures were kept in culture for 3 weeks (37 °C, 5% CO$_2$, and 95% humidity) before the start of the experiment to stabilize them after the cutting procedure. To induce oxidative stress, 0.25 mM sodium (meta) arsenite (Sigma) was added to the medium for 2 h. For crosslinking experiments, slices were subsequently scraped into PBS and a homogenous cell suspension was prepared by pipetting up and down. The cross-linking procedure was performed as described above with 1 mM DSG. Anti-TDP-43 antibody (Bethyl, Supplementary Table 1) was used for immunoblotting. For immunofluorescence imaging, slices were washed in PBS, fixed in 4% paraformaldehyde, blocked for 4 h with blocking buffer (10% donkey serum (DS), 0.3% Triton-X) and stained using anti-TDP-43 antibody (Proteintech, Supplementary Table 1), and anti-TIA-1 antibody (Santa Cruz, Supplementary Table 1) for 3 days. In all, 488- and 594-Alexa Fluor-conjugated secondary antibodies raised in donkey (Life technologies) were used (2 days incubation). Nuclei were stained with 4,6-diamidino-2-phenylindole (DAPI) and slices were mounted in fluorescent anti-fade mounting medium (Life technologies). Images were taken with an inverse confocal microscope (Leica DMI6000B, SP5) in the middle of the slice tissue.

**Protein expression and purification.** For bacterial recombinant protein expression, the DNA nucleotide sequence coding for full-length human TDP-43 (414 amino acids) was PCR amplified and cloned into the expression vector—pET-28a (+). Subsequently, the region corresponding to the NTD (amino acids 1–80) was sub-cloned into the pET28a(+) expression vector. Point mutations were incorporated by site-directed mutagenesis PCR using high fidelity Phusion DNA Polymerase (New England biolabs) with primers introducing DNA base substitutions (Supplementary Table 3). The protein was overexpressed in *Escherichia coli* host cells BL21 codon plus RIL strain (Agilent, Catalog number #230240). Bacterial cultures were induced at OD$_{600}$ = 1.0 by adding 1 mM isopropyl β-D-thiogalactoside (IPTG, Sigma) at 25 °C for 30 h in minimal M9 medium containing $^{15}NH_4Cl$ as nitrogen source and/or $^{13}C$-D-glucose as a carbon source for labeled protein samples. For unlabeled protein samples, bacterial cultures were grown in Luria-Bertani (LB) medium (Invitrogen). Following cell harvesting, the cell pellets were resuspended in lysis buffer that comprised of 50 mM sodium phosphate, pH 8.0, 300 mM NaCl, 10 mM imidazole, 1 mM dithiothreitol (DTT), 1 mM phenylmethylsulphonyl fluoride. Cell pellets were resuspended in 20 mL of lysis buffer per liter of bacterial culture pellet. The resuspended cells were then lysed by microfluidizer. Total cell lysates were then centrifuged (Sorvall SS-34 rotor, 45,000xg, 40 min). Pellet was discarded and the supernatant containing soluble protein was used for further purification. Protein purification was achieved on ÄKTA Prime purification system (Amersham Biosciences) where the N-terminal His-tagged protein was purified with Ni$^{2+}$-affinity chromatography under native conditions. The bound protein was eluted from Ni$^{2+}$-columns using a linear gradient of imidazole in elution buffer (50 mM sodium phosphate, pH 8.0, 1 M NaCl, 500 mM imidazole, 1 mM DTT). The fractions containing pure protein (analyzed on SDS polyacrylamide gel electrophoresis) were pooled and dialyzed overnight in a buffer without imidazole (5 mM sodium phosphate, pH 7.4, 1 mM NaCl, 1 mM DTT). Finally, the protein was concentrated using a Vivaspin 5000 MWCO concentrator (Sartorius Stedium Biotech). Further protein purification was achieved by size exclusion chromatography using a Superdex75 column (GE Healthcare). The purified protein was aliquoted and stored at −20 °C.

**Crystallization and data collection.** To screen for initial crystallization conditions, commercially available sparse matrix screens from Hampton Research (Hampton Research, California, USA) and Molecular Dimensions (Molecular Dimensions, Suffolk, UK) were used to set up sitting-drop vapor diffusion experiments. For this, purified protein at a concentration of 10 mg mL$^{-1}$ was mixed in a 3:1 (protein:mother liquor) ratio, dispersed on a 96 well Intelli-Plate 96-3 LVR crystallization plate (Art Robbins) using a Phoenix crystallization robot (Art Robbins) and incubated at 4 °C against 50 μL reservoir solution. An initial crystallization hit was found in 50 mM Tris/HCl pH 8.5, 8%(w/v) PEG 8000. These crystals were crushed and used for matrix microseeding in Crystal Screen 1&2 from Hampton Research (Hampton Research, California, USA). This led to new crystals in 0.1 M sodium cacodylate pH 6.5 and 1.4 M sodium acetate and a fine screen with differing pH and molarity of the sodium acetate was set-up, which resulted in crystals with good diffraction quality.

For data collection, crystals were incubated in mother liquor containing 30% ethylene glycol and flash-cooled in liquid nitrogen. Data were collected at beamline X06DA at the Swiss Light Source (PSI, Villigen, Switzerland) with a wavelength of 1 Å using the PILATUS 2M-F detector system (Dectris, Switzerland) and processed with XDS[46]. Due to the formation of weak ice-rings, data between 2.235 and 2.245 Å resolution were excluded. The outer resolution limit was estimated using the CC1/2[47] and I/σ(I) criteria[48]. According to these criteria diffraction data were cut at 2.1 Å resolution with CC1/2 and I/σ(I) of 59.2% and 1.6% in the outer resolution shell, respectively.

**Structure determination and refinement and analysis.** Using the anomalous signal from the dimethyl arsinic anion (cacodylate) linked to cysteine residues 39 and 50, phases and an initial poly-alanine model were obtained using the SHELX-C/D/E[49] programs in the software suite hkl2map[50]. An initial search in SHELX-D revealed four cacodylate ions and SHELX-E was used to build an initial poly-alanine trace with

20 cycles of density modification and a solvent content of 50%. After that, phenix autobuild[51, 52] was used to build the initial model. Further model building was done in COOT[53] and refinement was done with REFMAC5[54] in the CCP4-suite[55], phenix refine[56] and BUSTER (version 2.10.2. Cambridge, United Kingdom: Global Phasing Ltd). Table 1 shows the data processing and refinement parameters. Structure alignments of the two monomers in the asymmetric unit and external restraints were generated by ProSMART[57], buried surface area was calculated with PISA[58], and the interacting residues were drawn with Ligplot+[59]. To analyze the superhelical parameters of the structure we used the make_symmdef_file.pl script from the Rosetta Symmetry framework[60] in combination with the $C_\alpha$ model from the asymmetric unit. All structural figures were generated in Pymol[61].

**NMR spectroscopy.** The following NMR spectrometers were used for measuring the NMR experiments in this study—Bruker AVIII-500 MHz, AVIII-600 MHz, AVIII-700 MHz, and Avance-900 MHz that were equipped with cryoprobes. TopSpin2.1/TopSpin3.0 (Bruker) was used to perform data processing while the analysis was done in Sparky (http://www.cgl.ucsf.edu/home/sparky). For TDP-43 NTD backbone assignments, the following NMR experiments were measured (in H$_2$O)—two-dimensional (2D) $^{15}$N–$^1$H HSQC, 2D $^{13}$C–$^1$H HSQC, three-dimensional (3D) HNCA, 3D HNCOCA, 3D CACBCONH, 3D HCcH TOCSY, 3D hCCH TOCSY, 3D $^{15}$N-, and $^{13}$C-edited Nuclear Overhauser SpectroscopY (NOESYs).

**Electron microscopy.** Purified wild type and mutant TDP-43 NTD (100–500 μM) were applied to carbon-coated grids, stained with 2% (w/v) uranyl acetate and analyzed at various magnifications on 100 kV transmission electron microscope (Philips CM 100).

**Cell culture and transient transfections and immunostaining.** Mouse motor neuron-like hybrid cell line (NSC-34, Bioconcept, Catalog number—CLU140) were grown on cell culture dishes coated with matrigel in Dulbecco's modified Eagle's/F-12 medium (DMEM) supplemented with 10% fetal bovine serum, at 37 °C in an atmosphere of 5% CO$_2$. Mouse multipotent neural progenitor or stem-like C17.2 cells (Sigma, Catalog number—07062902) were grown on cell culture dishes in DMEM supplemented with 10% fetal bovine serum and 1% NEAA, at 37 °C in an atmosphere of 5% CO$_2$. Both NSC-34 and C17.2 cell lines were authenticated by the respective sources and were checked for Mycoplasma contamination in our laboratory.

For RNA splicing rescue experiments, transfections for transient over-expression of each vector (5 μg of DNA/2*10$^6$cells) was obtained with Lipofectamine 2000 (Life technologies) according to the manufacturer's instruction. After 12 h incubation with transfection reagents, the medium was replaced with normal growth medium. Depletion of endogenous mouse TDP-43 was performed by RNA interference as described previously[38]. The sequence of TDP-43 siRNA used is 5′-AGGAAUCAGCGUGCAUAUAUU-3′ and were used at 20 nM concentration. The sequence of control siRNA used is 5′-GUGCACAUGAGUGAGAUUU-3′.

For quantification of phosphorylated TDP-43 (pTDP-43), NSC-34 cells were plated on 24-well (ibidi) cell culture-plates at a density of ~5*10$^5$ cells per well leading to a confluency of approximately 80% at time of transfection. Prior to transfection, the cells were checked for confluency and morphology and the media was changed. Transfection was done with Lipofectamine 2000 transfection reagent according to the manufacturer's protocol and 0.25 μg of DNA per well was added. Cells were maintained at 37 °C under a humidified atmosphere of 5% CO$_2$ for 24 h before fixing them with 4% PFA for 10 min. For immunostaining, the cells were first permeabilized and blocked for 1 h with 10% DS and 0.2% Triton X-100 in PBS. Primary antibodies in PBS with 10% DS, 0.2% Triton X-100 were added and incubated for 2 h at RT. Cells were washed with PBS before incubating them with secondary antibodies for 1 h at RT and subsequently washed with PBS. After washing, coverslips were mounted with ProLong anti-fade medium with DAPI (Life technologies) and the plate was kept for at least 24 h in the dark before imaging. For the bipartite and tripartite GFP complementation assays, C17.2 cells were transiently transfected directly on poly-D-lysine-coated 8-well microscope slides with Lipofectamine 3000 and a total of 0.3 μg of DNA per well. Cells were fixed 24 h later with 4% PFA followed by treatment with 0.3% Triton X-100 and 5% normal goat serum in PBS. Immunostaining was performed with anti-human TDP-43 primary monoclonal antibody and secondary anti-mouse IgG Alexa594. Reconstituted GFP was directly visualized by fluorescence emission.

**RNA extraction and quantitative reverse transcription (RT)-PCR.** Total RNA from NSC-34 cells was obtained using TRIzol (Life Technologies) according to the manufacturer's instructions. Complementary DNA was generated from the total RNA (extracted from NSC-34 cells) using oligo-dT primers and Superscript III reverse transcriptase (Life technologies) as instructed by the manufacturer. Candidate TDP-43 splicing targets were tested by RT-PCR amplification using 24–27 cycles with specific primers (Supplementary Table 2) using Taq DNA polymerase (Thermo scientific). RT-PCR reaction products were analyzed on 10% polyacrylamide gels following staining with SYBR gold reagent (Life technologies). Quantification of distinct splice isoforms was done using ImageJ software (US National Institutes of Health). Finally, ratios of intensity between RT-PCR products with exon inclusion and exclusion were averaged for four biological replicates per condition.

**Sequential insolubility assay.** One 10 cm petri dish of cells transfected as described above was used for each sequential extraction of proteins with buffers of increasing stringency. Cells were harvested by scraping in solubilization buffer (SB: 10 mM Tris pH 7.4, 150 mM NaCl, 0.5 mM EDTA, 1 mM DTT, complete EDTA-free protease inhibitors (Roche). Total protein concentrations were measured and 1 mg of total protein was used for the SIA. Samples were treated with Benzonase (250 units per sample in 2 mM MgCl$_2$, 30 min at 4 °C (Roche)) before, and at each solubilization phase of SIA. For each step, samples were centrifuged for 30 min at 16,000×g at 4 °C. After removing the supernatant (soluble material), pellets (insoluble material) were suspended in SB 1% v/v Nonidet P40 Substitute (NP-40, Sigma), SB 2% w/v dodecyl-ß-D-maltoside (Sigma), SB 2% w/v N-Lauroylsarcosine (Sarkosyl, Sigma) and SB 1% w/v sodium dodecyl sulfate (SDS, Sigma) sequentially. SDS insoluble material was suspended in SB 1x SDS blue loading buffer (Life Technologies).

**Data availability.** Coordinates and structure factors have been deposited in the Protein Data Bank with accession code 5MDI. The chemical shift assignments for TDP-43 NTD have been deposited in the BioMagResBank (BMRB) under the accession number 27072. All other data are available from the corresponding author upon reasonable request.

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

## Acknowledgements

We thank Dr John Ravits (UCSD) for providing human brain tissue and Dr Shuo-Chien Ling (NUS) for providing GFP-tagged TDP-43 plasmids for mammalian cell expression. Dr Marian Hruska-Plochan for providing human fibroblasts, Dr Alexander Batyuk for help in structure determination, Sarah Erni for technical help with cell culture, Beat Blattmann and Céline Stutz-Ducommun (Protein crystallization center, University of Zürich) for help with protein crystallization screens, the staff from beamlines X06SA and X06DA from the Swiss Light Source (Paul Scherrer Institute, Villigen, Switzerland) for support during X-ray data collection, Gery Barmettler for technical help in TEM, Sonu Sahadevan, Katharina Hembach, Myriam Balerna, and Julia Lüdke for helpful technical advice and discussions. This work was supported by grants from the Swiss National Science Foundation (PP00P3_144862 to M.P. and 31003A_166612 to P.P.), and research grants from NCCR RNA and disease to M.P. and F.H.T.A. M.P. is the recipient of a Career Development Award from the Human Frontier Science Program and further acknowledges support from the Novartis research foundation.

## Author contributions

T.A. led the study and conducted in vitro and in vivo experiments. E.-M.H. performed in situ cross-linking in mouse slices. P.E., P.M., and T.A. measured the X-ray diffraction data and performed structure determination. P.M., T.A., and A.P. directed the structural analysis of the TDP-43 NTD. L.A.B.G. and F.L. performed transient transfections, imaging and quantifications of NSC-34 cells, M.J. performed RT-PCRs, Z.M. did cloning of TDP-43, C.F. and P.P. performed GFP complementation assay, T.A. and F.H.T.A. directed the NMR and biophysical experiments and M.P. directed the whole study. T.A., P.E., F.H.T.A. and M.P. drafted the manuscript. All authors read, edited and approved the final manuscript.

## Additional information

**Competing interests:** The authors declare no competing financial interests.

