## [Peer Review file · Nature Communications]

Reviewers' comments:

Reviewer #1 (Remarks to the Author):

TDP-43 has played a central role in ALS and FTD since cytoplasmic aggregates of this protein are prominent pathological features, particularly in ALS. In this manuscript, Polymenidou and colleagues investigate the structure and function of the TDP-43 80 aa N-terminal domain (NTD), which previous studies have proposed to be either unstructured, adopt a Ub-like fold or, more recently, a topological DIX domain-like fold. Using a membrane-permeable cross-linker (DSG), they first provide evidence for TDP-43 oligomers in human fibroblasts and cerebellum. This is followed by a detailed structural analysis of the NTD using a combination of crystal and NMR approaches, resulting in the discovery that the NTD forms head-to-tail solenoidal-like structures. Further evidence for NTD polymers is provided by EM visualization of NTD fibrillary ~7 nm structures. Point mutations predicted to disrupt this head-to-toe interface block oligomerization in vitro and in DSG-treated NSC-34 cells as monitored by GFP-tagged TDP-43. These NTD mutations also result in inhibition of TDP-43 regulated alternative splicing activity and a tripartite GFP complementation approach indicates that NTD-mediated TDP-43 oligomerization impairs LCD interactions. The authors conclude that the NTD mediates homo-oligomerization of TDP-43, which they propose is important for its nuclear RNA splicing role while also preventing deleterious LCD interactions that lead to aggregate formation and pathology. Overall, this is a novel and well-developed study that will contribute significantly to our understanding of the interrelationship between TDP-43 structure and function. Thus, I have only a few recommendations for additional controls and clarification.

1) Fig. 1a.

- a. The model in Fig. 7c indicates that NTD-mediated oligomerization is uniquely a nuclear event. However, the statement that higher MW TDP-43 bands are not apparent ('are detected only') in the C fraction begs the question of whether these higher MW species exist in the cytoplasm so this observation should be bolstered by a longer exposure of this fraction since the total C signal is much less than either the T or N signals.
- b. If the apparent (H3 signal is decreased but not GAPDH) increase in total TDP-43 at 500 μ M DSG is reproducible, the authors should provide an explanation why the peak exists at this cross-linker concentration.
- c. To support the uniqueness of this observation, it would be informative to show a comparable analysis for hnRNP A1 since it is structurally related to TDP-43, and has also been suggested to form functionally relevant oligomers, but does not have the NTD.
- d. The statement that the TDP-43 HMW species correspond to the distinct molecular masses of dimer, trimer and so on, requires further explanation since the monomer runs at ~50, the dimer at <80 and the trimer at <115 kDa.

2) Fig. 1b. Is the ratio of monomeric and higher MW TDP-43 similar between the fibroblasts and cerebellum (i.e., if the exposure time of the monomeric fibroblast signal is reduced to the cerebellum level, are the ratios in the two samples comparable)?

3) Fig. S1a&b. These are the same figures as in Fig. 1a&b but with the modest addition of MW markers. Since this also appears to be a lower exposure (at least S1a), both lower, and higher C, exposures could be used to address the points above.

4) Fig. 4f. What was the rationale for placing the GFP tag at the N-terminus adjacent to the NTD since GFP-TDP-43 appears to be restricted primarily to a dimer? Could this placement block the formation of larger oligomers?

5) Fig.5c. Please check the Sema3F splicing event. Which alternative exon was tested – 5 or 7?

6) Fig. S7, Results p10, ln278. The statement that wild type and mutant TDP-43 proteins are primarily nuclear in the majority of transfected cells is not supported by this figure – the majority transfected cells show a nuclear + cytoplasmic distribution pattern. Also, it is not clear from Fig. 7b and c if the confocal signals that overlap the nucleus are indeed true nuclear signals. Therefore, it would be informative to compare the relative GFP-TDP-43 and GFP-TDP-43-6M abundance in nuclear and cytoplasmic fractions by westerns.

Reviewer #2 (Remarks to the Author):

Referee Report for Afroz et al.

In this paper, the authors present a crystal structure of the N-terminal domain (NTD) of TDP-43, which, at the request of the editor, I looked at as a crystallographer/structural biologist. I am not an expert in cellular biology, so I cannot judge that part of the manuscript.

The NTD crystallizes with two molecules in the asymmetric unit, which through the P63 space group symmetry of the crystals results in three intertwined superhelices. The authors propose this superhelical arrangement to be physiologically relevant, and support this with EM, NMR and site-directed mutagenesis studies.

While the step from crystal structure to physiologically relevant oligomer is always tricky, the other data are indeed very supportive, although the EM images are very small on my printout and it is difficult to see much on them. I would probably have been a little bit more careful in my wording, and would have only said that the presented structure is proposed to be physiologically relevant. However, the authors greatly overestimate the resolution of their crystallographic data. Looking at the data statistics table, I am struck by the Rmerge and Rmeas values in the highest resolution shell. The authors cannot seriously mean to include data with an Rmerge of ~5 (i.e., ~500%)? Moreover, the CC1/2 is 0.21 in the highest resolution shell, which is far below reasonable standards, and the author's statement that the completeness shows their data to be of 1.95A resolution is unrealistic - that is no way to determine resolution.

So what happened here? At my request, the authors sent a pdb file and an mtz file. Unfortunately, the mtz did not list same unit cell as the pdb, which was the one that was reported in the paper. Moreover, the mtz file that was sent was indexed in a way that was incompatible with the pdb file. I would urge the authors to pay attention to stuff like this when they submit their pdb for deposition. Little mistakes of this kind look very bad indeed.

After correcting this, I looked at the density, which seems fine, but not really of <2 A resolution, as expected from the above considerations. Moreover, there are unusual bumps in the Wilson plot, at resolutions that seem very familiar. A plot of F vs resolution indeed shows strong peaks at exactly the resolutions expected for ice rings.

So I think the authors may have been fooled by the strong scattering of polycrystalline ice into thinking their data was of higher resolution than it really is. I estimate that their resolution is more like 2.5 A, but without the actual images it is hard to be sure.

However, and this is more important: the authors don't need better than 2 A resolution for what they want to do here. 2.5-3 A is absolutely fine for these purposes and I so would suggest that the authors look at their processing again, and then do a few more rounds of refinement against a dataset with a more realistic resolution cutoff.

Also, am I correct that the C-terminus of the NTD ends up on the inside of the superhelix? Is the 20 amino acids or so linker between the NTD and the next domain sufficient to allow the formation of the whole structure?

A minor point on line 100: immunoblotting should probably immunoblotting.

Moreover, I think that the stereo image in figure 2c is cross-eyed stereo (supplementary figure 2f is wall-eyed). I think it should be wall-eyed stereo throughout.

Reviewer #3 (Remarks to the Author):

The authors have performed an impressive array of crosslinking, crystallography, NMR and EM studies to show that the N-terminal domain of TDP-43 forms physiological oligomers. Based on the structural data, TDP-43 mutants were designed and tested for RNA splicing activity. The experimentally confirmed oligomerization-impaired TDP-43 mutants indeed showed less activity. This work provides very significant novel information about the structure-function relationship of TDP-43. The experiments were done carefully and are well controlled. The quality of the results is high and fully support the conclusions.

The last part where the authors claim that N-terminal oligomerization interferes with pathological aggregation is comparatively less convincing. The authors rely on only one end point, namely cytosolic phospho-TDP-43 immunostaining. The suggested enhancement of aggregation in the TDP-43 mutants might reach statistical significance, but overall the effect strength is low. More disturbingly, no inclusions are seen, only widespread cytosolic localization. What one may rather see is nucleocytoplasmic relocalization (the NLS is not far away from the studied N-terminal domain), which indeed could be a first step of pathological TDP-43 aggregation, but solid end points were not established here. The conclusions about pathological TDP-43 are better toned down and formulated as an outlook rather than formal proof by the present data.

The authors discuss a few mechanistic possibilities how oligomerization might affect RNA splicing activity. Apart from nuclear import and cytosolic phosphorylation (see above), direct RNA binding or conformational integrity of the catalytically active holoprotein could be affected. If time and efforts can be spent, direct RNA binding assays with the present TDP-43 mutants would be a good addition to this already great and impressive paper.

One little typo:

Figure 6g: significances **** and *** in legend *** and ** in graph. Please adjust.

Reviewer #4 (Remarks to the Author):

Afroz et al. have used crystallography to solve the atomic resolution structure of the N-terminal domain (NTD) of the protein TDP-43. TDP-43 is a multidomain protein containing a structured NTD, two RNA binding domains (RRM1 and RRM2), and an unstructured, low-complexity C-terminal domain (LCD) that is responsible for TDP-43 accumulation into stress granules and formation of cytoplasmic aggregates, which ultimately lead to disease development. The X-ray structure reported in this manuscript shows that the NTD can form oligomers through the interaction of a positively charged surface patch (head) on one monomeric subunit and a negatively charged surface patch (tail) on a second monomeric subunit. The authors proved the existence of these oligomers in solution by NMR and electron microscopy. At this point the authors set out to understand the physiological relevance of NTD oligomerization *in vivo*. They showed that mutations in the head and tail regions of NTD inhibit DSG-crosslinking of the full-length TDP-43 *in vivo*, and that these mutants are not able to rescue splicing defects induced by depletion of endogenous TDP-43. The authors conclude that TDP-43 oligomerizes *in vivo* through its N-terminal domain, and that this oligomerization is important for the correct TDP-43 functioning in RNA metabolisms. Finally, they used the tripartite split GFP technology to investigate the proximity of the NTD's and LCD's of different TDP-43 molecules in the oligomer *in vivo*. Tagging of the NTD results in fluorescent samples, while tagged LCD's fail to reconstitute fluorescence, indicating that the NTDs of adjoining TDP-43 molecules are in close proximity, while LCD's are distantly located from each other. The authors hypothesize that TDP-43 oligomerization through the NTD can antagonize cytoplasmic aggregation through LCD, and show that overexpression of the oligomerization defective mutants increases (slightly) the amount of cytoplasmic aggregates in NSC-34 cells.

Overall the manuscript is well written and the experiments are up-to-date and carefully performed. My only concern is that the Tm and Hm mutants completely abolish oligomerization in vitro, but have no effect on oligomerization in vivo (not sure if I got this right because in vivo data are not shown for these mutants). How do the authors explain this finding? Can they report the in vivo cross-linking data for these mutants? Is it possible that the head or tail regions are involved in a different interaction (i.e. not the NTD oligomerization) in vivo that would induce colocalization of several TDP-43?

Other minor concerns:

EM shows that the NTD oligomers form ~7nm filaments, but the NMR spectra are very well resolved. How do the authors reconcile these two findings? Is it possible that they have a range of exchanging oligomeric species in solution? It would be very informative to determine the oligomeric state of NTD by AUC or similar technique.

Following transient overexpression of TDP-43, NSC-34 cells develop cytoplasmic aggregates.

Overexpression of the oligomerization-defective mutants results in a slightly higher accumulation of cytoplasmic aggregates. Can you exclude that the expression level of the oligomerization defective mutants is higher than WT TDP-43? Can this be the cause of your result?

Reviewer #1 (Remarks to the Author):

TDP-43 has played a central role in ALS and FTD since cytoplasmic aggregates of this protein are prominent pathological features, particularly in ALS. In this manuscript, Polymenidou and colleagues investigate the structure and function of the TDP-43 80 aa N-terminal domain (NTD), which previous studies have proposed to be either unstructured, adopt a Ub-like fold or, more recently, a topological DIX domain-like fold. Using a membrane-permeable cross-linker (DSG), they first provide evidence for TDP-43 oligomers in human fibroblasts and cerebellum. This is followed by a detailed structural analysis of the NTD using a combination of crystal and NMR approaches, resulting in the discovery that the NTD forms head-to-tail solenoidal-like structures. Further evidence for NTD polymers is provided by EM visualization of NTD fibrillary ~7 nm structures. Point mutations predicted to disrupt this head-to-toe interface block oligomerization in vitro and in DSG-treated NSC-34 cells as monitored by GFP-tagged TDP-43. These NTD mutations also result in inhibition of TDP-43 regulated alternative splicing activity and a tripartite GFP complementation approach indicates that NTD-mediated TDP-43 oligomerization impairs LCD interactions. The authors conclude that the NTD mediates homo-oligomerization of TDP-43, which they propose is important for its nuclear RNA splicing role while also preventing deleterious LCD interactions that lead to aggregate formation and pathology. Overall, this is a novel and well-developed study that will contribute significantly to our understanding of the interrelationship between TDP-43 structure and function. Thus, I have only a few recommendations for additional controls and clarification.

Author comment: We gratefully acknowledge this referee's assessment that "this is a novel and well-developed study that will contribute significantly to our understanding of the interrelationship between TDP-43 structure and function".

1) Fig. 1a.

a. The model in Fig. 7c indicates that NTD-mediated oligomerization is uniquely a nuclear event. However, the statement that higher MW TDP-43 bands are not apparent ('are detected only') in the C fraction begs the question of whether these higher MW species exist in the cytoplasm so this observation should be bolstered by a longer exposure of this fraction since the total C signal is much less than either the T or N signals.

Author response: We thank the referee for raising this important point. In our new **Supplementary Figures 1a and 2d**, we have now included longer exposures of the blots shown in the main Figure 1a and 1e, respectively, as recommended by the reviewer. Consistent with previous reports, under physiological conditions, the majority (~99%) of total TDP-43 is localized to the nucleus. Therefore, it is indeed not surprising that the vast majority of the physiological oligomers are found in the nucleus (**Fig. 1a**). Upon high exposure, very low levels of oligomeric TDP-43 species are detectable in the cytosolic fraction (new **Supplementary Fig. 1a**), but given their low abundance, we cannot exclude the possibility that these originate from a nuclear leak during the nucleo-cytoplasmic fractionation.

TDP-43 is a nucleo-cytoplasmic shuttling protein and has roles in both the nucleus and the cytoplasm^{1,2}. In order to increase the cytoplasmic fraction of TDP-43, we have used oxidative stress conditions that lead to TDP-43 incorporation within cytoplasmic stress granules. Despite detectable increase of cytoplasmic TDP-43 by immunofluorescence (**Fig. 1d**), we did not observe any significant increase in the total oligomer-to-monomer ratio (new **Supplementary Fig. 2e**). Upon high exposure of the blot, we do detect some oligomeric species in the cytosolic fraction as well (new **Supplementary Fig. 2d**), but these too, do not seem significantly increased with cellular stress.

Based on the observation that cellular stress does not significantly alter the level or localization of normal oligomers, we propose that the majority of NTD-mediated TDP-43 oligomers are nuclear, as shown in the scheme of **Fig. 7c**. This may be due to the fact that high local concentration of TDP-43 in nucleus

would potentially favor TDP-43 oligomerization, in agreement with our NMR monitored concentration titration of TDP-43 NTD (**Fig. 3a**). Nevertheless, we cannot exclude the presence of TDP-43 oligomers in high protein-RNA concentration sites in cytoplasm, such as stress granules. Our data, however, suggest that the stress granule-associated TDP-43 complexes may be structurally distinct from the NTD-mediated oligomers. This is in line with previous observations suggesting that the low complexity domain of TDP-43 is mediating its association with stress granules³. Ongoing work in our lab is aimed at clarifying the relationship between NTD-driven oligomerization of TDP-43 with its stress granule incorporation, but this requires a considerable amount of work, which will be the focus of another manuscript.

b. If the apparent (H3 signal is decreased but not GAPDH) increase in total TDP-43 at 500 μ M DSG is reproducible, the authors should provide an explanation why the peak exists at this cross-linker concentration.

Author response: To address this point we quantified the dimer to monomer ratio of several independent crosslinking experiments, with increasing DSG concentrations. The data that are shown in our new **Supplementary Fig. 1c** indicate that this peak represents experimental variability, potentially due to high viscosity of total (T) and nuclear (N) fractions caused by genomic DNA. Overall, we do see increased intensity of high molecular weight TDP-43 bands upon increasing DSG concentration, without a consistent peak in one specific concentration.

c. To support the uniqueness of this observation, it would be informative to show a comparable analysis for hnRNP A1 since it is structurally related to TDP-43, and has also been suggested to form functionally relevant oligomers, but does not have the NTD.

Author response: As suggested by the reviewer, we performed cross-linking in human fibroblasts followed by nucleo-cytoplasmic separation for two other hnRNP family of proteins – hnRNP A1 and FUS. These data are now included in our new **Supplementary Fig. 2a**. hnRNP A1 has been proposed to form dimers via the glycine rich domain⁴ and further oligomerize upon binding to RNA⁵. Following incubation with increasing DSG concentrations, we detected a specific dimer band of hnRNP A1 migrating at 76 kDa (**Supplementary Fig. 2a, upper panel**), but no higher bands at sizes equivalent to trimer, tetramer, etc, as we see for TDP-43. Instead, we obtain very high molecular weight hnRNP A1-positive complexes that are unable to enter the 4-12% polyacrylamide gels, potentially representing larger complexes of hnRNP A1 with RNA. Following a similar cross-linking approach for FUS, we only observed very high molecular weight species that were not able to enter the gels (**Supplementary Fig. 2a, second panel**).

hnRNP A1 and FUS are similar to TDP-43 harboring tandem RRM domains and a low complexity domain. However, they lack the N-terminal domain that mediates TDP-43 oligomerization in a specific head-to-tail fashion. In contrast, hnRNP A1⁵ potentially utilizes its low-complexity domain for dimerization and RNA binding seems to be essential for its oligomerization. However, the structural basis of this oligomerization remains to be elucidated. In any case, these additional experiments give us confidence that the NTD-mediated oligomerization of TDP-43 is indeed unique and we thank the referee for suggesting them.

d. The statement that the TDP-43 HMW species correspond to the distinct molecular masses of dimer, trimer and so on, requires further explanation since the monomer runs at ~50, the dimer at <80 and the trimer at <115 kDa.

Author response: We thank the referee for noticing this slight discrepancy in the marker position in immunoblot in **Fig. 1a**, which happened during figure formatting and which we have now corrected. To keep clarity of all gel migrations, we have included unformatted versions of all our western blots and

DNA gels in our original supplementary figures, in which the molecular markers are also visible. Based on these sizes, we are convinced that the observed HMW species observed upon cross-linking indeed represent dimers, trimers and so on. We have repeatedly observed that the electrophoretic migration pattern of TDP-43 is slightly different (a little higher) than the predicted size of 43 kDa, consistent with published literature⁶⁻⁸. Also, as is the case for many other proteins in total cell lysates, we also find that TDP-43 migration varies, depending on the kind of gel used for analysis. This explains the different migration of TDP-43 on 12% (**Fig. 1b**) compared to 4-12% denaturing polyacrylamide gels (**Fig. 1a**).

2) Fig. 1b. Is the ratio of monomeric and higher MW TDP-43 similar between the fibroblasts and cerebellum (i.e., if the exposure time of the monomeric fibroblast signal is reduced to the cerebellum level, are the ratios in the two samples comparable)?

Author response: This is an important question that we have addressed by quantifying the ratio of monomeric to dimeric TDP-43 of several independent crosslinking experiments and across various biological samples reported in this study – i.e. human fibroblasts, human cerebellum, human motor cortex and mouse brain slices. These ratios are quite comparable as seen in our new **Supplementary Fig. 1f**, especially considering the level of complexity of various samples analyzed.

3) Fig. S1a&b. These are the same figures as in Fig. 1a&b but with the modest addition of MW markers. Since this also appears to be a lower exposure (at least S1a), both lower, and higher C, exposures could be used to address the points above.

Author response: We included **Fig. S1a & b** with markers to be transparent in the presentation of our western blots in the main figures. As explained in point d above, this is particularly important in this case. As recommended by the

reviewer, we have included both higher and lower exposures of the immunoblot to address the first point.

4) *Fig. 4f. What was the rationale for placing the GFP tag at the N-terminus adjacent to the NTD since GFP-TDP-43 appears to be restricted primarily to a dimer? Could this placement block the formation of larger oligomers?*

Author response: At its C-terminal side TDP-43 has a low complexity domain, which is not predicted to adopt much secondary structure and is a protein-protein interaction domain⁹. We wanted to leave this domain unperturbed and therefore decided to place GFP with a long flexible linker at the N-terminus of TDP-43. The N-terminally tagged protein is fully functional, since it can efficiently rescue the splicing defect induced by down-regulation of endogenous TDP-43, as seen in the RNA splicing assays (**Fig. 5c**). Based on this functionality and on the fact that the linker between GFP and TDP-43 is long and flexible, we do not think that this causes a block in the oligomer formation. Instead, we think that the reason that the high molecular weight TDP-43 species, beyond dimers, were not obvious in the immunoblot of our original **Fig. 4f** is their size. Indeed, since the combined size of the monomeric GFP-TDP-43 fusion protein is 77 kDa, its dimer would be 154 kDa, which is still visible on the blot. However, its trimer and tetramer would correspond to 231 or 308 kDa respectively, which are not resolved distinctly in the standard immunoblot. To better resolve these oligomer species, we have used 4-12% polyacrylamide gels with MOPS running buffer, and allowed the gel to migrate long, until sizes 40 kDa reached the end of the gel. This blot, which is now included as our new main **Fig. 4f**, clearly demonstrates that the GFP-TDP-43 fusion protein forms not only dimers, but also trimers. Therefore, we are fully convinced that GFP placing on the N-terminal side of TDP-43 does not interfere with its oligomerization.

5) *Fig.5c. Please check the Sema3F splicing event. Which alternative exon was tested – 5 or 7?*

Author response: We have corrected this in **Fig. 5c** where we tested splicing of alternative exon 5.

6) *Fig. S7, Results p10, ln278. The statement that wild type and mutant TDP-43 proteins are primarily nuclear in the majority of transfected cells is not supported by this figure – the majority transfected cells show a nuclear + cytoplasmic distribution pattern. Also, it is not clear from Fig. 7b and c if the confocal signals that overlap the nucleus are indeed true nuclear signals. Therefore, it would be informative to compare the relative GFP-TDP-43 and GFP-TDP-43-6M abundance in nuclear and cytoplasmic fractions by westerns.*

Author response: We followed the referee's suggestion and performed nucleocytoplasmic fractionation of NSC-34 cells transiently transfected with either wild type or oligomerization mutant TDP-43. We have included this in our new **Supplementary Fig. S8f-g**. Consistent with immunofluorescence, we see both nuclear and cytoplasmic localization of TDP-43 in both wild type and oligomerization mutants. Quantification of the nuclear to cytoplasmic ratio from three independent experiments showed a slight increase in cytoplasmic TDP-43 in the oligomerization-deficient mutant, which, however, similarly to immunofluorescence quantifications, did not reach statistical significance (new **Supplementary Fig. S8g**).

Reviewer #2 (Remarks to the Author):

Referee Report for Afroz et al.

In this paper, the authors present a crystal structure of the N-terminal domain (NTD) of TDP-43, which, at the request of the editor, I looked at as a crystallographer/structural biologist. I am not an expert in cellular biology, so I cannot judge that part of the manuscript.

The NTD crystallizes with two molecules in the asymmetric unit, which through the P63 space group symmetry of the crystals results in three intertwined superhelices. The authors propose this superhelical arrangement to be physiologically relevant, and support this with EM, NMR and site-directed mutagenesis studies.

While the step from crystal structure to physiologically relevant oligomer is always tricky, the other data are indeed very supportive, although the EM images are very small on my printout and it is difficult to see much on them. I would probably have been a little bit more careful in my wording, and would have only said that the presented structure is proposed to be physiologically relevant.

However, the authors greatly overestimate the resolution of their crystallographic data. Looking at the data statistics table, I am struck by the Rmerge and Rmeas values in the highest resolution shell. The authors cannot seriously mean to include data with an Rmerge of ~5 (i.e., ~500%)? Moreover, the CC1/2 is 0.21 in the highest resolution shell, which is far below reasonable standards, and the author's statement that the completeness shows their data to be of 1.95A resolution is unrealistic - that is no way to determine resolution.

So what happened here? At my request, the authors sent a pdb file and an mtz file. Unfortunately, the mtz did not list same unit cell as the pdb, which was the one that was reported in the paper. Moreover, the mtz file that was sent was indexed in a way that was incompatible with the pdb file.

I would urge the authors to pay attention to stuff like this when they submit their pdb for deposition. Little mistakes of this kind look very bad indeed.

After correcting this, I looked at the density, which seems fine, but not really of <2 A resolution, as expected from the above considerations. Moreover, there are unusual bumps in the Wilson plot, at resolutions that seem very familiar. A plot of F vs resolution indeed shows strong peaks at exactly the resolutions expected for ice rings.

So I think the authors may have been fooled by the strong scattering of polycrystalline ice into thinking their data was of higher resolution than it really is.

I estimate that their resolution is more like 2.5 Å, but without the actual images it is hard to be sure.

However, and this is more important: the authors don't need better than 2 Å resolution for what they want to do here. 2.5-3 Å is absolutely fine for these purposes and I so would suggest that the authors look at their processing again, and then do a few more rounds of refinement against a dataset with a more realistic resolution cutoff.

Author response: We are grateful to the reviewer for his/her detailed evaluation of our structural work and the very helpful comments. We apologize for sending out the wrong combination of MTZ and PDB-files for review. We accidentally sent an older version of the MTZ file, which was indeed incompatible with the PDB file. The deposited files were of course consistent in terms of unit cell parameters and indexing.

We would particularly like to thank the reviewer for taking the time to look at the Wilson plot, which shows narrow spikes at 2.24 Å and 2.06 Å resolution. Re-inspection of the raw data showed no sign of continuous ice rings at lower resolution, but the higher resolution data has been affected by the superposition of reflections from the protein with reflections from a few ice crystals. We reprocessed the data as suggested and excised the affected resolution shells. This diminished the overall completeness from 99.9 to 98.8%, but had no effect on $I/\sigma(I)$ or CC1/2. The refinement of the structure against the reprocessed data reduced the free R-factor of the structure from 24.3 to 20.6%. As already anticipated by the reviewer, the additional refinement rounds had no effect on the interpretation of the structure or any other findings discussed in the manuscript.

The reviewer suggests that the resolution should be truncated at 2.5 Å because Rmerge in the highest resolution shell is too high. The proper way to determine the resolution cutoff is indeed discussed controversially in the scientific community. Following the suggestions of Karplus and Diederichs^{10,11} we would like to cut the reprocessed data at $CC1/2 > 50\%$ and $I/\sigma(I) > 1$, which results in a resolution cutoff of 2.1 Å resolution. As suggested by Karplus and Diederichs,

Rmeas “*should play no role in determining [the] resolution cutoff*”. We concur with this opinion, because Rmeas is defined as the intensity error divided by the intensity. Because the intensity approaches zero at high resolution, Rmeas must go to infinity. At 2.5 Å resolution the reprocessed data possess $CC1/2 = 0.95$ and $I/\sigma(I) = 5.6$, which is much higher than even the most conservative estimates and confirms that the resolution range between 2.5 and 2.1 Å contains valuable information. Although we do not make use of this information in the discussion of the structure, as the reviewer also indicates, this information would have been lost if the data is cut at 2.5 Å resolution.

To satisfy the recommendations of the reviewer we have truncated the data at 2.1 Å resolution and we have updated **Table 1** accordingly. We also modified the statement how the resolution cutoff was determined and we have included the appropriate reference for it.

Also, am I correct that the C-terminus of the NTD ends up on the inside of the superhelix? Is the 20 amino acids or so linker between the NTD and the next domain sufficient to allow the formation of the whole structure?

Author response: The C-terminus of the NTD ends up in different planes with respect to consecutive molecules in one superhelix in the crystal structure. Indeed, one can model the rest of protein domains in the superhelix without steric clashes. At the moment, we are trying to obtain additional experimental data to generate a structural model of full-length protein using modular domain structure reported in this and in previous studies. This, however, will be presented in a future manuscript.

A minor point on line 100: immunoblotting should probably immunoblotting. Moreover, I think that the stereo image in figure 2c is cross-eyed stereo (supplementary figure 2f is wall-eyed). I think it should be wall-eyed stereo throughout.

Author response: We have corrected in line 100 to “immunoblotting” and changed the **Fig. 2c** to be wall-eyed stereo.

Reviewer #3 (Remarks to the Author):

The authors have performed an impressive array of crosslinking, crystallography, NMR and EM studies to show that the N-terminal domain of TDP-43 forms physiological oligomers. Based on the structural data, TDP-43 mutants were designed and tested for RNA splicing activity. The experimentally confirmed oligomerization-impaired TDP-43 mutants indeed showed less activity. This work provides very significant novel information about the structure-function relationship of TDP-43. The experiments were done carefully and are well controlled. The quality of the results is high and fully support the conclusions.

Author comment: We thank the referee for the very positive evaluation of our study and his/her constructive comments.

The last part where the authors claim that N-terminal oligomerization interferes with pathological aggregation is comparatively less convincing. The authors rely on only one end point, namely cytosolic phospho-TDP-43 immunostaining. The suggested enhancement of aggregation in the TDP-43 mutants might reach statistical significance, but overall the effect strength is low. More disturbingly, no inclusions are seen, only widespread cytosolic localization.

Author response: The referee raises an important point that we have now investigated in more depth. In fact the idea that physiologic oligomerization may interfere with pathological aggregation was initially inspired by the GFP complementation experiments, which showed that the low complexity domains of TDP-43 – whose tight interaction is key for aggregation – are not in close proximity within the physiological oligomers. We measured the presence of cytosolic phosphorylated TDP-43, because we have repeatedly observed that

this phosphorylation (at the C-terminus) indeed marks pathologic TDP-43 assemblies and is essentially undetectable in normal human or mouse brain, in accordance with published work^{12,13}. Even if the presence of phosphorylated TDP-43 does not equal biochemical aggregation, its presence does indicate the initiation of aggregation. To further monitor the aggregation propensity of oligomerization mutants, we now provide biochemical evidence of aggregation, in our new **Supplementary Fig. 13a-c**. It was previously shown that pathologic TDP-43 in ALS and FTD patients becomes abnormally insoluble to strong detergents, such as Sarkosyl¹². Using sequential insolubility assay from cell lysates, we observed a significant increase in the sarkosyl-insoluble fraction of oligomerization mutants compared to wild type TDP-43, consistent with our initial claim.

What one may rather see is nucleocytoplasmic relocalization (the NLS is not far away from the studied N-terminal domain), which indeed could be a first step of pathological TDP-43 aggregation, but solid end points were not established here. The conclusions about pathological TDP-43 are better toned down and formulated as an outlook rather than formal proof by the present data.

Author response: We agree with the reviewer that since NLS (a.a. 82-84) is located in proximity to the NTD, it is conceivable that our mutations could lead to re-localization of TDP-43 that might be the first step toward cytoplasmic aggregation. We have systematically quantified the nuclear and cytoplasmic localization of our oligomerization mutants and compared it to the wild-type protein by immunofluorescence (**Supplementary Fig. S8a-d**) and by nucleocytoplasmic fractionation (**Supplementary Fig. S8f-g**). Both assays showed a slight increase in cytoplasmic TDP-43 in the oligomerization-deficient mutant, which, however, did not reach statistical significance (please also see above – referee #1, point 6). While we cannot exclude that this slight increase may contribute to the higher TDP-43 phosphorylation and aggregation that we

observed in oligomerization mutant-expressing cells, we think that this alone cannot fully explain our findings.

The authors discuss a few mechanistic possibilities how oligomerization might affect RNA splicing activity. Apart from nuclear import and cytosolic phosphorylation (see above), direct RNA binding or conformational integrity of the catalytically active holoprotein could be affected. If time and efforts can be spent, direct RNA binding assays with the present TDP-43 mutants would be a good addition to this already great and impressive paper.

Author response: The reviewer is addressing the mechanistic basis of the functional role of TDP-43 oligomerization in RNA metabolism, which is indeed a very important question. In order to answer the first question of whether the oligomerization mutations interfere with the RNA binding activity of TDP-43 holoprotein, we have first excluded that our mutations unfold the N-terminal domain itself (**Supplementary Fig. 6d**). This strongly suggests that our mutations do not interfere with TDP-43 binding to RNA via the tandem RNA recognition motifs (RRM1 and RRM2), which occurs in a sequence-specific fashion (to UG-rich RNA)¹⁴. As a further support of this conclusion, we now provide evidence that mutations within the NTD, which are independent of the oligomerization interface are fully functional on our splicing assay, strongly suggesting that RNA-binding is unaffected in this case (new **Supplementary Fig. 9a-c**). What we cannot exclude by these assays is that the NTD itself directly binds RNA, which may also be influenced by oligomerization mutations. To address this, we performed gel shift assay of the isolated NTD protein to TG-repeats and we found a very weak association of this domain with this nucleic acid sequence (new **Supplementary Fig. 14a**). This is consistent with published work showing that TDP-43 RRMs are indispensable for sequence-specific RNA recognition and the N-terminal domain contributes only marginally to the overall RNA binding affinity of TDP-43¹⁵.

One little typo:

Figure 6g: significances **** and *** in legend *** and ** in graph. Please adjust.

Author response: We have corrected this in the graph and the legend.

Reviewer #4 (Remarks to the Author):

Afroz et al. have used crystallography to solve the atomic resolution structure of the N-terminal domain (NTD) of the protein TDP-43. TDP-43 is a multidomain protein containing a structured NTD, two RNA binding domains (RRM1 and RRM2), and an unstructured, low-complexity C-terminal domain (LCD) that is responsible for TDP-43 accumulation into stress granules and formation of cytoplasmic aggregates, which ultimately lead to disease development. The X-ray structure reported in this manuscript shows that the NTD can form oligomers through the interaction of a positively charged surface patch (head) on one monomeric subunit and a negatively charged surface patch (tail) on a second monomeric subunit. The authors proved the existence of these oligomers in solution by NMR and electron microscopy. At this point the authors set out to understand the physiological relevance of NTD oligomerization in vivo. They showed that mutations in the head and tail regions of NTD inhibit DSG-crosslinking of the full-length TDP-43 in vivo, and that these mutants are not able to rescue splicing defects induced by depletion of endogenous TDP-43. The authors conclude that TDP-43 oligomerizes in vivo through its N-terminal domain, and that this oligomerization is important for the correct TDP-43 functioning in RNA metabolisms. Finally, they used the tripartite split GFP technology to investigate the proximity of the NTD's and LCD's of different TDP-43 molecules in the oligomer in vivo. Tagging of the NTD results in fluorescent samples, while tagged LCD's fail to reconstitute fluorescence, indicating that the NTDs of adjoining TDP-43 molecules are in close proximity, while LCD's are distantly located from each other. The authors hypothesize that TDP-43 oligomerization through the NTD can antagonize cytoplasmic aggregation through LCD, and

show that overexpression of the oligomerization defective mutants increases (slightly) the amount of cytoplasmic aggregates in NSC-34 cells. Overall the manuscript is well written and the experiments are up-to-date and carefully performed.

Author comment: We gratefully acknowledge the referee's statement that "Overall the manuscript is well written and the experiments are up-to-date and carefully performed".

My only concern is that the Tm and Hm mutants completely abolish oligomerization in vitro, but have no effect on oligomerization in vivo (not sure if I got this right because in vivo data are not shown for these mutants). How do the authors explain this finding? Can they report the in vivo cross-linking data for these mutants? Is it possible that the head or tail regions are involved in a different interaction (i.e. not the NTD oligomerization) in vivo that would induce co-localization of several TDP-43?

Author response: We thank the reviewer for addressing this important comment. Based on the electrostatic inter-molecular interaction interface seen in the crystal structure, neutralization of one charged surface should be sufficient to abolish inter-molecular interaction. This is seen *in vitro* in NMR and EM images, where Hm (R52AR55A) or Tm (E17AE21A) fail to oligomerize (**Supplementary Fig. 7a-d**). Following the referee's suggestion, we have now included both the *in vivo* oligomerization (new **Fig. 4f**) and splicing data for these mutants as well (new **Supplementary Fig. 9d**). In our new **Fig. 4f**, we show that both Hm and Tm mutants show significantly lower oligomerization ability in cells. In line with this, we now report that mutation of only one interface (data reported for Tm) is sufficient to abolish the function of TDP-43 in splicing (new **Supplementary Fig. 9d**). We therefore conclude that interfering with one charged interface (just two point mutations E17AE21A) is sufficient to block inter-molecular interactions and the function of protein *in vivo*.

Other minor concerns:

EM shows that the NTD oligomers form ~7nm filaments, but the NMR spectra are very well resolved. How do the authors reconcile these two findings? Is it possible that they have a range of exchanging oligomeric species in solution? It would be very informative to determine the oligomeric state of NTD by AUC or similar technique.

Author response: In case of TDP-43 NTD, the NMR spectra show an average of fast exchanging states. The increase in protein concentration not only results in specific chemical shift perturbations but also significant line broadening suggesting the formation of high molecular weight species in a fast exchange regime on NMR time scale. Indeed, the fast exchange regime leads to averaging of line widths between monomeric and oligomeric protein states as seen in the ^1H - ^{15}N HSQC spectrum. Indeed, EM images corroborate this hypothesis where in addition to long fibrils, smaller oligomers are seen as well. However, due to low affinity of interaction (low micromolar), these mixed populations were not resolved distinctly in size exclusion chromatography, where the oligomers potentially disassemble while passing through the column. In order to systematically analyze the *in vitro* state of full-length protein, we are working on the characterization of the size by analytical ultracentrifugation as recommended by the reviewer. However, this requires substantial optimization, especially because of the high aggregation nature of the full-length protein. We agree with the reviewer that this is a minor point that would not significantly influence the main conclusions of the current work, so we will include these results in a follow-up manuscript.

Following transient overexpression of TDP-43, NSC-34 cells develop cytoplasmic aggregates. Overexpression of the oligomerization-defective mutants results in a slightly higher accumulation of cytoplasmic aggregates. Can you exclude that the

expression level of the oligomerization defective mutants is higher than WT TDP-43? Can this be the cause of your result?

Author response: To address this point, we confirmed biochemically on immunoblots the expression levels of wild type and various oligomerization mutants. We have included this data in **Supplementary Fig. 8e**, where we see that the expression levels of wild type protein are similar to that of oligomerization mutants. So, we do think that the increase in the percentage of cells with phosphorylated cytoplasmic aggregates is due to their inability to form dynamic physiological oligomers. Furthermore, a similar increase in phosphorylated TDP-43 aggregates upon deletion of the entire N-terminal domain necessary for oligomerization corroborates the result. Additionally, the oligomerization mutant proteins show increased insolubility compared to the wild type protein in the sequential insolubility assays (new **Supplementary Fig. 13a-d**). Please also see above referee#3, point #1.

References -

- 1 Buratti, E. & Baralle, F. E. Characterization and functional implications of the RNA binding properties of nuclear factor TDP-43, a novel splicing regulator of CFTR exon 9. *J Biol Chem* 276, 36337-36343, doi:10.1074/jbc.M104236200 (2001).
- 2 Polymenidou, M. *et al.* Long pre-mRNA depletion and RNA missplicing contribute to neuronal vulnerability from loss of TDP-43. *Nat Neurosci* 14, 459-468, doi:10.1038/nn.2779 (2011).
- 3 Dewey, C. M. *et al.* TDP-43 Is Directed to Stress Granules by Sorbitol, a Novel Physiological Osmotic and Oxidative Stressor. *J Bacteriol* 193, 1098-U1108, doi:10.1128/Mcb.01279-10 (2011).
- 4 Cartegni, L. *et al.* hnRNP A1 selectively interacts through its Gly-rich domain with different RNA-binding proteins. *J Mol Biol* 259, 337-348 (1996).
- 5 Okunola, H. L. & Krainer, A. R. Cooperative-binding and splicing-repressive properties of hnRNP A1. *Mol Cell Biol* 29, 5620-5631, doi:10.1128/MCB.01678-08 (2009).
- 6 Cohen, T. J., Hwang, A. W., Unger, T., Trojanowski, J. Q. & Lee, V. M. Redox signalling directly regulates TDP-43 via cysteine oxidation and disulphide cross-linking. *EMBO J* 31, 1241-1252, doi:10.1038/emboj.2011.471 (2012).
- 7 Bozzo, F. *et al.* Structural insights into the multi-determinant aggregation of TDP-43 in motor neuron-like cells. *Neurobiol Dis* 94, 63-72 (2016).
- 8 Swarup, V. *et al.* Deregulation of TDP-43 in amyotrophic lateral sclerosis triggers nuclear factor kappaB-mediated pathogenic pathways. *J Exp Med* 208, 2429-2447 (2011).
- 9 Buratti, E. *et al.* TDP-43 binds heterogeneous nuclear ribonucleoprotein A/B through its C-terminal tail: an important region for the inhibition of cystic fibrosis transmembrane conductance regulator exon 9 splicing. *The Journal of biological chemistry* 280, 37572-37584 (2005).
- 10 Karplus, P. A. & Diederichs, K. Linking crystallographic model and data quality. *Science* 336, 1030-1033, doi:10.1126/science.1218231 (2012).
- 11 Karplus, P. A. & Diederichs, K. Assessing and maximizing data quality in macromolecular crystallography. *Curr Opin Struct Biol* 34, 60-68, doi:10.1016/j.sbi.2015.07.003 (2015).
- 12 Neumann, M. *et al.* Ubiquitinated TDP-43 in frontotemporal lobar degeneration and amyotrophic lateral sclerosis. *Science* 314, 130-133, doi:10.1126/science.1134108 (2006).
- 13 Lagier-Tourenne, C. *et al.* Divergent roles of ALS-linked proteins FUS/TLS and TDP-43 intersect in processing long pre-mRNAs. *Nat Neurosci* 15, 1488-1497, doi:10.1038/nn.3230 (2012).
- 14 Lukavsky, P. J. *et al.* Molecular basis of UG-rich RNA recognition by the human splicing factor TDP-43. *Nat Struct Mol Biol* 20, 1443-1449, doi:10.1038/nsmb.2698 (2013).
- 15 Chang, C. K. *et al.* The N-terminus of TDP-43 promotes its oligomerization and enhances DNA binding affinity. *Biochem Biophys Res Co* 425, 219-224, doi:10.1016/j.bbrc.2012.07.071 (2012).

REVIEWERS' COMMENTS:

Reviewer #1 (Remarks to the Author):

The authors have responded to all of my comments and I have no further concerns.

Reviewer #2 (Remarks to the Author):

I am pleased to say that the authors have taken away all my concerns. I can therefore wholeheartedly recommend publication of this manuscript.

Reviewer #3 (Remarks to the Author):

The authors have further improved their manuscript, including biochemical evidence for initial aggregation of mutant TDP-43 (new supplementary figure 13), systematic assessment of nucleocytoplasmic distribution (new supplementary figure 8fg), demonstration of equal protein expression levels, etc.

I feel the work is ready for publication.

Just a little detail: there is no need to amend supplementary figure 12 (a) as there are no additional subfigure panels.

Reviewer #4 (Remarks to the Author):

The authors have addressed all my comments/concerns. I think the manuscript is suitable for publication in Nature Communications.